# A genome-wide CRISPR screen identifies WDFY3 as a regulator of macrophage efferocytosis

Jianting Shi [1,14], Xun Wu[1,14], Ziyi Wang[1], Fang Li[1], Yujiao Meng[1,2], Rebecca M. Moore[1], Jian Cui[1], Chenyi Xue[1], Katherine R. Croce[3], Arif Yurdagul Jr[4], John G. Doench [5], Wei Li[6,7], Konstantinos S. Zarbalis [8,9,10], Ira Tabas[3,11,12], Ai Yamamoto[3,13] & Hanrui Zhang [1] ✉

Phagocytic clearance of dying cells, termed efferocytosis, is essential for maintaining tissue homeostasis, yet our understanding of efferocytosis regulation remains incomplete. Here we perform a FACS-based, genome-wide CRISPR knockout screen in primary mouse macrophages to search for novel regulators of efferocytosis. The results show that *Wdfy3* knockout in macrophages specifically impairs uptake, but not binding, of apoptotic cells due to defective actin disassembly. Additionally, WDFY3 interacts with GABARAP, thus facilitating LC3 lipidation and subsequent lysosomal acidification to permit the degradation of apoptotic cell components. Mechanistically, while the C-terminus of WDFY3 is sufficient to rescue the impaired degradation induced by *Wdfy3* knockout, full-length WDFY3 is required to reconstitute the uptake of apoptotic cells. Finally, WDFY3 is also required for efficient efferocytosis in vivo in mice and in vitro in primary human macrophages. This work thus expands our knowledge of the mechanisms of macrophage efferocytosis, as well as supports genome-wide CRISPR screen as a platform for interrogating complex functional phenotypes in primary macrophages.

Phagocytic clearance of dead or dying cells by phagocytes, a process known as efferocytosis, is important in embryogenesis and development, and the resolution of pathological events[1–4]. Impaired efferocytosis lessens the effective clearance of dying cells, causing secondary necrotic cell death and damages[1–4]. Efferocytosis is performed by macrophages and to a lesser extent by other professional phagocytes (such as monocytes and

dendritic cells), non-professional phagocytes and specialized phagocytes[1]. Because of the fundamental role of efferocytosis, dysregulation of this process is associated with many pathological states, including autoimmune diseases, atherosclerosis, and cancers[2]. Given the importance of this biological process and the therapeutic potential of targeting genes regulating efferocytosis, identifying novel regulators and mechanisms of this biological

[1]Cardiometabolic Genomics Program, Division of Cardiology, Department of Medicine, Columbia University Irving Medical Center, New York, NY, USA. [2]Beijing University of Chinese Medicine, Beijing, China. [3]Department of Pathology and Cell Biology, Columbia University, New York, NY, USA. [4]Department of Molecular & Cellular Physiology, Louisiana State University Health Sciences Center at Shreveport, Shreveport, LA, USA. [5]Broad Institute of MIT and Harvard, Cambridge, MA, USA. [6]Center for Genetic Medicine Research, Children's National Hospital, Washington, DC, USA. [7]Department of Genomics and Precision Medicine, George Washington University, Washington, DC, USA. [8]University of California at Davis, Department of Pathology and Laboratory Medicine, Sacramento, CA 95817, USA. [9]Shriners Hospitals for Children Northern California, Sacramento, CA 95817, USA. [10]UC Davis MIND Institute, Sacramento, CA 95817, USA. [11]Department of Medicine, Columbia University, New York, NY, USA. [12]Department of Physiology and Cellular Biophysics, Columbia University, New York, NY, USA. [13]Department of Neurology, Columbia University, New York, NY, USA. [14]These authors contributed equally: Jianting Shi, Xun Wu. ✉ e-mail: hz2418@cumc.columbia.edu

process has broad impacts on many diseases relevant to defective efferocytosis[5–8].

Hypothesis-driven approaches have successfully identified many key regulators for the removal of dying cells via efferocytosis[1–4]. Yet, an unbiased approach to screening regulators of efferocytosis of apoptotic cells (ACs) on a genome-wide scale is lacking. Unbiased screenings allow the identification of new regulators from diverse and unexpected gene classes. Genetic screens of efferocytosis of ACs have been performed in Drosophila[9], but not in mammalian cells. In mammalian cells, genome-wide CRISPR knockout screens have identified regulators of diverse substrates in differentiated myeloid leukemia cells[10,11] and macrophage-like cells[12], illuminating both universal and specific principles of phagocytosis, but not of ACs. However, a

screening platform using ACs as the substrates and in primary macrophages is critical because efferocytosis involves AC-specific recognition receptors[13], stiffness and size-dependent engulfment mechanisms[14], and cellular response to degradation[4], all of which cannot be recapitulated by phagocytosis of beads. In addition, immortalized or tumor-derived monocytic cell lines often lack physiological relevance to resemble fully the spectrum of physiological function in primary macrophages[15].

To address this gap, we establish and perform a pooled genome-wide CRISPR knockout screen for efferocytosis in primary murine bone marrow-derived macrophages (BMDMs) derived from the *Rosa26-Cas9* knock-in mice constitutively expressing Cas9 endonuclease. Our screen successfully identifies well-known key regulators

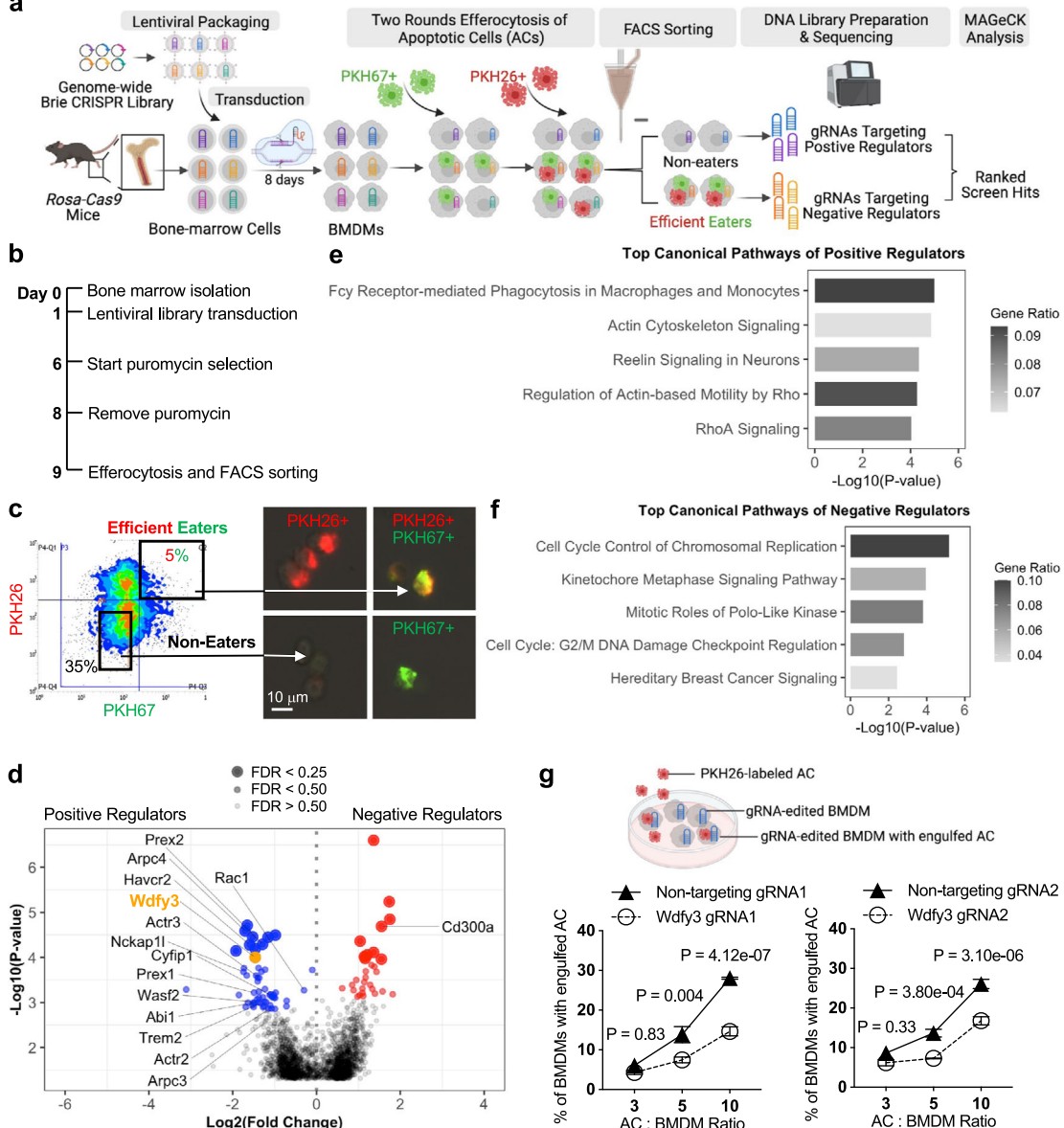

**Fig. 1 | A pooled, fluorescence-activated cell sorting (FACS)-based genome-wide CRISPR knockout screen in primary mouse macrophages identified known and novel regulators of macrophage efferocytosis. a** Schematics of the CRISPR screen workflow. **b** Timeline of bone marrow isolation, lentiviral library transduction, puromycin selection, efferocytosis, and cell sorting. **c** Visualization of gating strategy for separation of non-eaters and efficient eaters. Successful separation was confirmed by fluorescent microscopy. **d** Volcano plot highlights the top-ranked screen hits that are known positive and negative regulators of macrophage

efferocytosis. **e** Canonical pathways enriched in top-ranked positive regulators by Ingenuity Pathway Analysis (IPA). **f** Canonical pathways enriched in top-ranked negative regulators by IPA. **g** Validation of *Wdfy3* as a positive regulator required for macrophage efferocytosis (*n* = 4 independent experiments). Data are presented as mean ± SEM. Two-sided *P* values were determined by a two-way ANOVA with Tukey's multiple comparisons test in panel **g**. ACs, apoptotic cells; BMDMs, bone-marrow-derived macrophages.

responsible for the recognition and uptake of ACs, supporting the screen's performance. Individual validation of the strongest hits uncovers WDFY3 (WD repeat and FYVE domain containing 3), also known as Alfy (Autophagy-linked FYVE Protein), as a novel regulator previously not implicated in the regulation of efferocytosis or phagocytosis. We further uncover the novel mechanisms by which WDFY3 regulates the uptake and degradation of ACs during efferocytosis and demonstrate the role of WDFY3-mediated efferocytosis in vivo in mice and in vitro in primary human macrophages. Our study also establishes a broadly applicable platform for the genome-wide screen of complex functional phenotypes in primary macrophages for unbiased novel discoveries.

## Results

### A pooled, fluorescence-activated cell sorting (FACS)-based genome-wide CRISPR knockout screen in primary mouse macrophages identified known and novel regulators of macrophage efferocytosis

Genome-wide forward genetic screens have the capacity to examine a biological process in an unbiased manner and allow for novel discoveries. We first determined the proper cell types for a genome-wide CRISPR screen of macrophage efferocytosis. Human monocytic cell lines, including U937 and THP-1, can be differentiated to macrophage-like cells, which have previously been used for genome-wide screening[10,11]. Yet, we confirmed that U937 and THP-1 derived macrophages were not a proper model for screening of efferocytosis as monocytic cell line-derived macrophages showed poor efferocytosis capacity. Specifically, upon up to 24 h of AC incubation, only less than 1-3 % of either U937-derived or THP-1-derived macrophages were able to engulf ACs (Supplementary Fig. 1a). The results highlight the importance of using physiologically-relevant primary macrophages for screening of efferocytosis regulators.

We thus leveraged the *Rosa26-Cas9* knock-in mice constitutively expressing Cas9 endonuclease[16] and established a workflow for CRISPR gene editing in primary BMDMs. Specifically, lentiviral gRNA libraries were transduced to isolated bone marrow (BM) cells, which were then differentiated to BMDMs using L cell-conditioned media that provide macrophage colony-stimulating factor (M-CSF) for macrophage differentiation. As illustrated in Fig. 1a and Fig. 1b, for each replicate, 400-500 million BM cells were isolated and seeded. The lentiviral Brie library[17] (Addgene 73633) including 78,637 gRNAs targeting 19,674 mouse genes and 1000 non-targeting control gRNAs was transduced on day 1 with a low Multiplicity of Infection (MOI) to ensure that majority of the BM cells integrate one viral particle for gene editing of a single gene (the transduction rate is shown in Supplementary Fig. 1b). 48 h after transduction, puromycin was applied to select BM cells with successful lentiviral integration.

The success of the screening relies on the effective enrichment of macrophages with high vs. low efferocytosis capacity. Since efferocytosis is a binary event, to facilitate an effective separation and enrichment, we performed two rounds of efferocytosis sequentially. Specifically, human Jurkat cells (~10 μm in diameter), an acute T cell leukemia cell line routinely used for in vitro efferocytosis assays, were treated with staurosporine to induce apoptosis, then labeled with fluorescent linkers, PKH67 (Ex/Em: 490/502 nm) or PKH26 (Ex/Em: 551/567 nm), that stains cell membrane. BMDMs were first incubated with PKH67-labeled ACs at a ratio of 5:1 for AC: BMDM and allowed for efferocytosis. After 45 min, the unbound PKH67-labeled ACs were washed away and BMDMs were cultured for 2 h without ACs to allow degradation of the engulfed cargo. Next, BMDMs were fed with PKH26-labeled ACs also at a ratio of 5:1. After 90 min, unbound ACs were washed away and BMDMs were collected for flow cytometry sorting. Longer time was allowed for the second round in order to enrich BMDMs that engulf a second AC. Sorting separated the BMDMs that engulfed both PKH67+ and PKH26+ ACs, i.e., the efficient eaters (~5%),

and BMDMs that did not engulf any ACs, i.e., the non-eaters (Fig. 1c and Supplementary Fig. 1b). Two independent replicates were performed (Supplementary Fig. 1c). For each replicate, efferocytosis was performed in ~80 million BMDMs on day 9 (Supplementary Fig. 1b). After sorting, we obtained ~3 million efficient eaters and ~16 million non-eaters. We have also collected 40 million BMDMs on day 9 without performing efferocytosis, i.e. the input samples (Supplementary Fig. 1b).

We sequenced the sorted non-eaters, efficient eaters, and the input samples for each of the two replicates and performed MAGeCK analysis[18–21] to identify the top hits. We analyzed three comparisons: input vs. non-eaters (Supplementary Data 1), input vs. efficient eaters (Supplementary Data 2), non-eaters vs. efficient eaters (Supplementary Data 3). We expect that the comparison of input vs. non-eaters will identify positive regulators whose knockout impairs efferocytosis, while the comparison of input vs. efficient eaters will identify negative regulators whose knockout enhances efferocytosis. The comparison of non-eaters vs. efficient eaters likely further improves the power to identify enriched gRNAs. As expected, the analysis comparing non-eaters vs. efficient eaters was able to identify more known regulators (Fig. 1d, Supplementary Data 3 for the complete MAGeCK output). Non-targeting gRNAs did not show enrichment in either sample (Supplementary Fig. 1d).

The non-eaters are expected to be enriched for gRNAs targeting positive regulators essential for efferocytosis, i.e., knockout would impair efferocytosis. Indeed, we identified many genes involved in actin polymerization that is known to be essential for phagocytic cup formation, including *Rac1*, four members of the five-subunit SCAR/WAVE complex (*Nckap1l*, *Wasf2*, *Abi1*, *Cyfip1*) and five members of the seven-subunit ARP2/3 complex (*Actr2*, *Actr3*, *Arpc3*, and *Arpc4*) (Fig. 1d). We performed pathway analysis using Ingenuity Pathway Analysis (IPA). The top-ranked positive regulators (negative score <0.002, 163 genes) were enriched for pathways including Fcγ Receptor-mediated Phagocytosis in Macrophages and Monocytes, Actin Cytoskeleton Signaling etc. (Fig. 1e and Supplementary Data 4), supporting the screening performance in identifying well-known positive regulators. The results also show that many, but not all, genes involved in actin cytoskeleton remodeling and general phagocytosis are among the most highly ranked screen hits (Supplementary Fig. 2).

Using high-content imaging analysis, we selectively validated *Arpc4* (top-2 ranked) and *Nckap1l* (top-14 ranked) using the gRNAs from the original screening library. gRNAs targeting *Arpc4* or *Nckap1l* led to ~50% reduction in the efferocytosis of PKH26-labeled ACs by BMDM (Supplementary Fig. 1e). *Hacvr2*, also known as TIM3, is one of the PtdSer-specific receptors involved in AC recognition and efferocytosis[22]. *Hacvr2* was ranked at top-7 and was also validated with ~30% reduction in efferocytosis capacity (Supplementary Fig. 1e).

The efficient eaters are expected to be enriched for gRNAs targeting negative regulators, i.e. knockout would enhance efferocytosis. Efferocytosis needs to be tightly controlled and there are very few known negative regulators. While this manuscript is being prepared, the top-2 ranked hit for negative regulators, *Cd300a* (Fig. 1d), was identified as a novel negative regulator[23]. Specifically, the binding of an AC with Cd300a and the activation of downstream signaling suppresses efferocytosis by myeloid cells, thus the blockage of Cd300a enhanced efferocytosis[23]. We were also able to validate the results in BMDM using a gRNA targeting *Cd300a* (Supplementary Fig. 1e). Pathway analysis of top-ranked negative regulators (positive score <0.001 for a total of 96 genes) implies that genes involved in cell cycle control and chromosomal replication were enriched for top hits for negative regulators (Fig. 1f and Supplementary Data 5).

The screen has revealed many top-ranked hits that promise to inform novel biology and warrant further validation and functional interrogation. Among the top hits for positive regulators, *Wdfy3* is top-

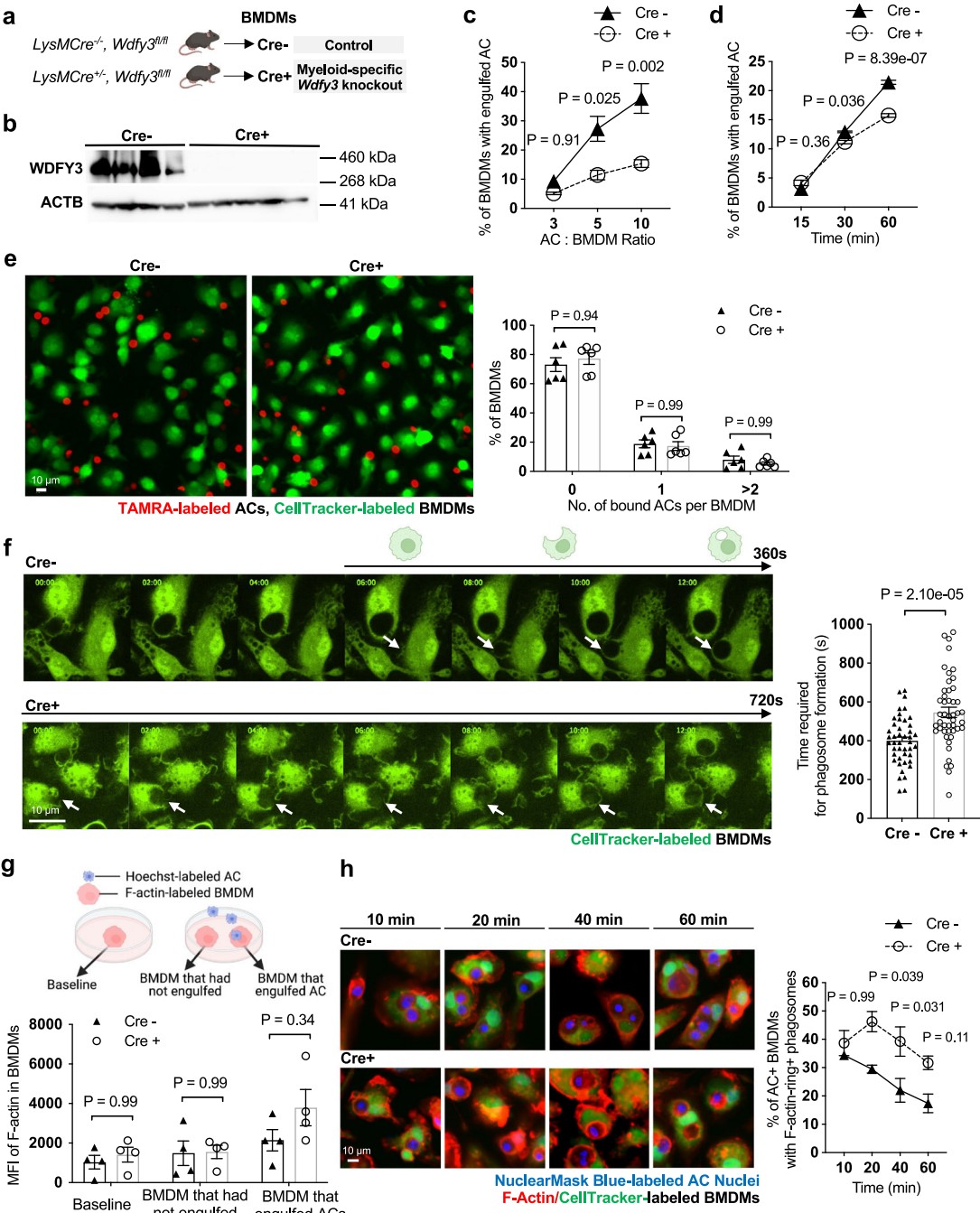

**Fig. 2 | WDFY3 deficiency led to impaired uptake, as opposed to binding, of apoptotic cells (ACs) due to defective actin depolymerization. a** Schematics of breeding *LysMCre* mice with *Wdfy3*$^{fl/fl}$ mice to obtain mice with myeloid-specific knockout of *Wdfy3*. **b** Validation of efficient knockout in BMDMs by western blot of WDFY3 (*n* = 4 biological replicates; the blot shown is a representative image of three independent experiments). **c** Cre⁻ and Cre⁺ BMDMs were incubated with Hoechst-labeled ACs at various AC: BMDM ratios of 3:1, 5:1, 10:1 respectively for 1 h and analyzed by flow cytometry (*n* = 3 biological replicates, each from the average of 2 technical replicates). **d** Cre⁻ and Cre⁺ BMDMs were incubated with PKH26-labeled ACs at various time points of 15 min, 30 min, and 60 min at a AC: BMDM ratio of 5:1 and analyzed by flow cytometry (*n* = 3 technical replicates). **e** Cre⁻ and Cre⁺ BMDMs were pretreated with cytochalasin D for 30 min to block poly-merization and elongation of actin, thus testing the binding of ACs with BMDMs. The treated BMDMs were then incubated with TAMRA-stained apoptotic mouse thymocytes at 37 °C for 30 min and then extensively washed with DPBS to remove unbound ACs for imaging and quantification after fixation (*n* = 6 biological repli-cates). **f** Cre⁻ and Cre⁺ BMDMs were stained with CellTracker and incubated with

ACs. Efferocytosis of ACs by BMDMs were observed using time-lapse confocal microscopy. The time required for phagosome formation was recorded and quantified (*n* = 44 and 47 data points for Cre- and Cre+ respectively, each data point represents one BMDM with engulfed ACs, 4 biological replicates for each genotype). The arrows point to the BMDM engulfing an AC across the stages from phagocytic cup formation to phagosome closure (from left to right). **g** F-actin labeled by siR-actin in Cre⁻ and Cre⁺ BMDMs was quantified by flow cytometry (*n* = 4 biological replicates, each from the average of 3 technical replicates). **h** BMDMs were stained with CellTracker and siR-actin, then incubated with NuclearMask Blue-labeled apoptotic Jurkat cells for various time points (10 min, 20 min, 40 min, and 60 min). For each time point, unbound ACs were removed and BMDMs were fixed. BMDMs were imaged and the percentage of BMDMs with engulfed cargos sur-rounded by F-actin rings in all BMDMs with engulfed cargos was quantified (*n* = 4 biological replicates, data are representative of two independent experiments). Data are presented as mean ± SEM. Two-sided *P* values were determined by a two-way ANOVA with Tukey's multiple comparisons test in (**c**, **d**, **e**, **g**, **h**), or by unpaired *t* test in panel **f**.

10 ranked but not previously implicated in the regulation of efferocytosis or phagocytosis, nor identified in previous screens in non-mammalian cells or using other substrates (Supplementary Fig. 2 and Supplementary Data 6). Using two individual gRNAs, one from the Brie library and one designed independently, and quantitative imaging analysis, we validated that knockout of *Wdfy3* in BMDMs led to impaired efferocytosis of PKH26-labeled ACs (Fig. 1g) without affecting BMDM viability (Supplementary Fig. 1f). The defects were more significant when BMDMs were challenged with higher AC to BMDM ratio, i.e., a condition mimicking high-burden efferocytosis.

Altogether, we established a CRISPR screen for regulators of efferocytosis, a complex functional phenotype, in primary macrophages at genome-wide coverage. Moreover, using this screen we uncovered a novel regulator, *Wdfy3*.

## WDFY3 deficiency led to impaired uptake, as opposed to binding, of apoptotic cells due to defective actin depolymerization

*WDFY3* encodes a highly conserved, large 400 kDa protein with 3526 amino acids. Similarly to mouse[24], *WDFY3* mRNA is the most abundantly expressed in the brain (Supplementary Fig. 3a), and in multiple brain cell types (Supplementary Fig. 3b). Among immune cells, *WDFY3* mRNA expression is abundant in myeloid cells, including macrophages, neutrophils, and monocytes, but not T cells (Supplementary Fig. 3b and Supplementary Fig. 3c).

To further validate the role of *Wdfy3* knockout in efferocytosis ex vivo, we obtained *Wdfy3*^fl/fl mice created by insertion of two loxP sites flanking exon 5 on a 129/SvEv x C57BL/6 background (generated by the Ai Yamamoto lab[24]). Myeloid-specific *Wdfy3* null mice were generated by breeding *Wdfy3*^fl/fl mice with *LysMCre* mice (JAX 004781), i.e. *LysMCre*^+/- *Wdfy3*^fl/fl mice (Cre^+) while using *LysMCre*^-/- *Wdfy3*^fl/fl littermates (Cre^-) as the controls (as illustrated in Fig. 2a). We confirmed efficient knockout by western blot of WDFY3 in BMDMs from the Cre^+ mice (Fig. 2b). Although global deletion of *Wdfy3* led to perinatal lethality[24], myeloid-specific loss of *Wdfy3* did not affect body weight (Supplementary Fig. 4a) or organ weight, including that of heart, liver, and spleen (Supplementary Fig. 4b). Moreover, the mice did not show changes in circulating levels of neutrophils and monocytes, confirming that myelopoiesis was not affected (Supplementary Fig. 4c).

We used flow cytometry to quantify the percentage of BMDMs with engulfed ACs labeled with Hoechst. With lower AC: BMDM ratio or at relatively early time points, efferocytosis of Cre^- and Cre^+ BMDMs appeared similar (Fig. 2c, d). The defects were more significant with a high ratio of AC: BMDM that resembles high-burden efferocytosis (Fig. 2c). Consistently, with an AC: BMDM ratio at 5:1, the defective efferocytosis in Cre^+ BMDMs was the most significant at later time points (Fig. 2d), also supporting more pronounced defects over prolonged periods of challenges. These effects were independent of sex (Supplementary Fig. 5a).

Efferocytosis involves the finding, recognition and binding, uptake, and finally the degradation of the engulfed cargos[1,2]. Our screen identifies regulators essential for the binding and/or uptake of ACs, but is not designed to identify genes involved solely in cargo degradation, if the binding or uptake of ACs is unaffected. The screen will also not identify genes solely responsible for the chemotactic cues termed find-me signals because the pooled design masks the defective secretion by a small subset of edited cells.

With this in mind, we next set out to determine the molecular steps regulated by WDFY3. We first aimed at determining if *Wdfy3* knockout affected binding and/or uptake during efferocytosis. TAMRA-labeled apoptotic murine thymocytes were incubated with CellTracker-labeled BMDMs pretreated with cytochalasin D that prevents actin polymerization thus the uptake of ACs at a 5:1 AC: BMDM ratio[25–27]. Following incubation, unbound ACs were washed away and BMDMs were fixed and imaged. The numbers of TAMRA-labeled ACs bound with each BMDM were counted and the percentage of BMDMs with none, one, or two and more bound ACs was quantified for Cre^- and Cre^+ BMDMs (the quantification strategy for binding is illustrated in Supplementary Note 1). The results support that *Wdfy3* knockout did not affect the ability of BMDMs to bind ACs (Fig. 2e), suggesting that the uptake, as opposed to binding, of ACs was impaired due to *Wdfy3* knockout.

Time-lapse live-cell imaging confirmed that the time required for complete internalization of ACs was longer in *Wdfy3* knockout BMDM compared with control (Fig. 2f), suggesting delayed phagosome formation. Phagosome formation during phagocytosis of large particles requires the coordination of actin polymerization and depolymerization, permitting the continual restructuring of the actin cytoskeleton[14]. Complete internalization of the cargo is synchronized with actin depolymerization, allowing subsequent phagosome maturation[28] (as also visualized in Supplementary Movie 1). We thus asked if *Wdfy3* knockout affects actin polymerization and/or depolymerization. We labeled BMDMs with siR-actin, a fluorogenic, cell-permeable probe based on an F-actin binding natural product jasplakinolide, and determined F-actin levels at baseline and upon efferocytosis of Hoechst-labeled ACs by flow cytometry. F-actin signals at baseline were similar between Cre^- and Cre^+ BMDMs (Fig. 2g, left panel). Upon efferocytosis, BMDMs that had not engulfed ACs also showed comparable F-actin levels between Cre^- and Cre^+ BMDMs (Fig. 2g, middle panel). Yet, in BMDMs that engulfed ACs, *Wdfy3* knockout BMDMs showed a trend of higher F-actin signals (Fig. 2g, right panel, $P = 0.34$). Although not statistically significant, the observed trend led us to hypothesize that potential defects in actin disassembly exist in *Wdfy3* knockout BMDMs. Indeed, in *Wdfy3* knockout BMDMs that had successfully internalized an AC, we observed that many engulfed ACs were surrounded by F-actin rings (Fig. 2h). Confirming our subjective observations, the percentage of BMDMs with F-actin surrounded cargos over all BMDMs that had engulfed ACs was greater in Cre^+ vs. Cre^- BMDMs (Fig. 2h). In Cre^- control BMDMs, the percentage of BMDMs with F-actin surrounded cargos was the highest at 10 min after adding ACs and then decreased over time (Fig. 2h). Yet, in Cre^+ BMDMs, the percentage further increased and peaked at 20 min after adding ACs and remain greater than the percentage in Cre^- BMDMs (Fig. 2h), supporting defective actin depolymerization (the quantification strategy for F-actin rings is illustrated in Supplementary Note 2).

Thus, defective actin depolymerization in *Wdfy3* knockout macrophages led to impaired uptake and delayed phagosome formation during efferocytosis. The defects were specific to efferocytosis of ACs because the phagocytosis of other substrates, including polystyrene beads of different sizes (4 μm and 10 μm, Supplementary Fig. 5b and Supplementary Fig. 5c), sheep red blood cells (RBCs) that were untreated, stressed by heat treatment, or IgG-opsonized (Supplementary Fig. 5d), zymosan particles (500 nm, Supplementary Fig. 5e), was not impaired in *Wdfy3* knockout BMDM. Consistently, previous screens using the above-mentioned substrates in U937 monocytic line-derived macrophages[11], or using cancer cells in J774 macrophages[12] did not uncover *Wdfy3* as a hit (Supplementary Fig. 2 and Supplementary Data 6). Thus, we discovered and validated a novel regulator specifically required for the uptake of ACs during efferocytosis.

How macrophages involve different molecular machinery to regulate the engulfment of various cargos remains largely unknown. Recent studies revealed that the engulfment of larger cargos (e.g. 5 μm beads) requires phosphoinositide 3-kinase (PI3K)-mediated PtdIns(3,4,5)_3 production and PtdIns(3,4,5)_3-dependent recruitment of GTPase-activating proteins (GAPs) that inactivates Rho GTPases Rac/Cdc42, therefore allowing cycling of F-actin assembly and disassembly[14]. We expect that this mechanism is also required for the engulfment of ACs (~10 μm for Jurkat cells). Indeed, PI3K inhibitor, LY294002, markedly reduced the uptake of ACs in both control and *Wdfy3* knockout BMDMs (Supplementary Fig. 5f), implicating that WDFY3 was not required for PI3K activation. We reasoned that if

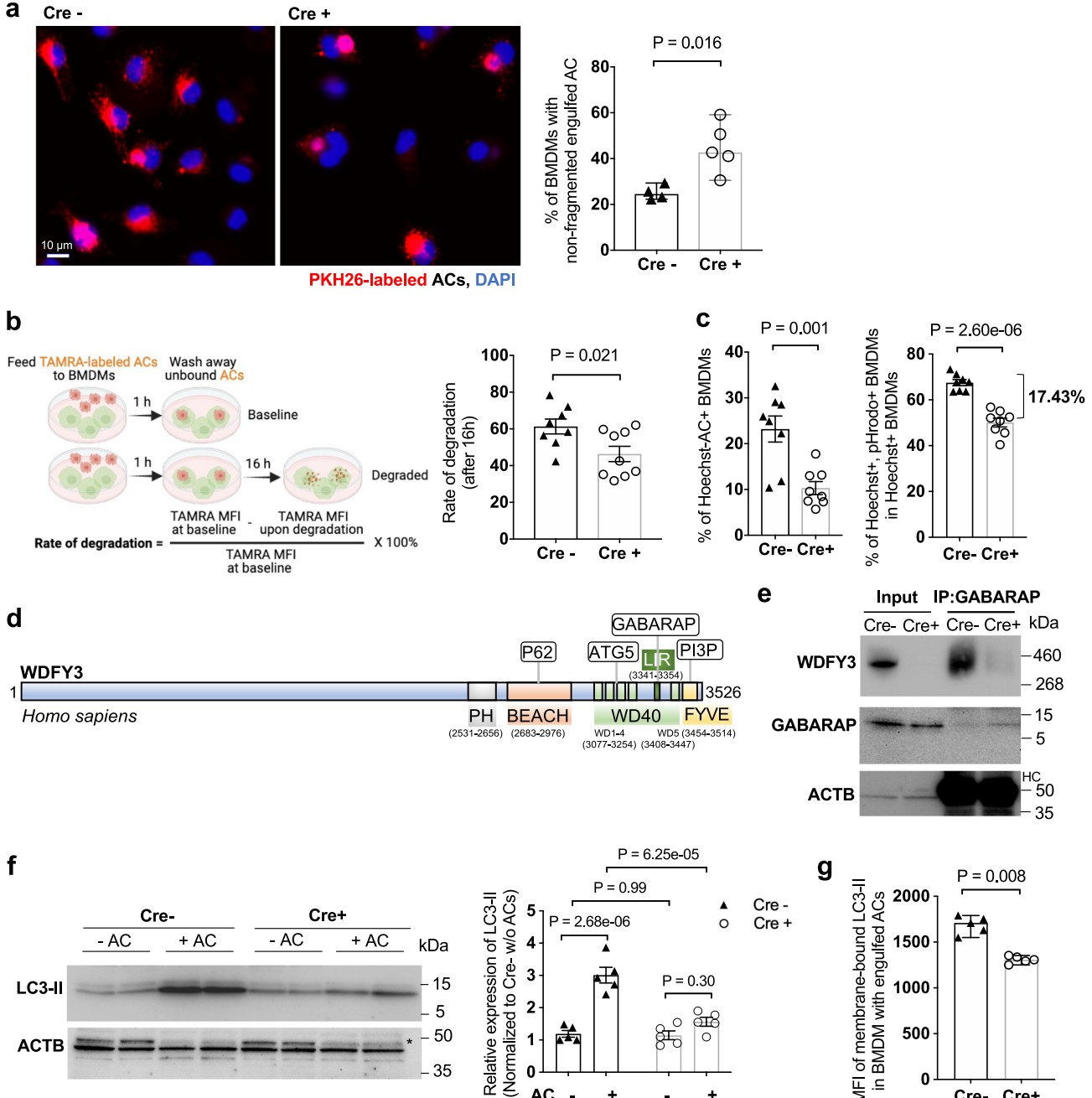

**Fig. 3 | WDFY3 deficiency led to defects in LC3-associated phagocytosis (LAP) and the degradation of engulfed ACs. a** Cre⁻ and Cre⁺ BMDMs were incubated with PKH26-labeled ACs for 1 h. After washing away the unengulfed ACs, BMDMs were placed back to the incubator for another 3 h. BMDMs were then fixed and imaged. The percentage of BMDMs showing non-fragmented PKH26 signals in the total number of PKH26⁺ BMDMs was quantified (n = 4 and 5 biological replicates for Cre− and Cre+ respectively, each from the average of 3 technical replicates). **b** Cre⁻ and Cre⁺ BMDMs were incubated with TAMRA-labeled ACs for 1 h. After washing away the unengulfed ACs, BMDMs were either collected for flow cytometry to quantify the MFI of TAMRA or placed back to the incubator for another 16 h and then collected for flow cytometry. The rate of degradation was calculated, as shown in the schematics (n = 8 and 9 biological replicates for Cre− and Cre+ respectively). **c** Cre⁻ and Cre⁺ BMDMs were incubated with ACs labeled by Hoechst, which stains DNA and is pH-insensitive, and pHrodo, which is pH-sensitive and shows fluorescent signal only under an acidified environment in the phagolysosome. The percentage of Hoechst⁺ BMDMs indicates uptake. The percentage of Hoechst⁺/pHrodo⁺ BMDMs in Hoechst⁺ BMDMs indicates acidification of the engulfed cargos (n = 8 biological replicates, each from the average of 2 technical

replicates). **d** Schematics of known functional domains and binding partners of human WDFY3. **e** The interaction between WDFY3 and GABARAP was assessed by co-immunoprecipitation. Cre⁻ and Cre⁺ BMDM cell lysates were incubated with anti-GABARAP antibody and protein A/G agarose beads. Beads-bound proteins were detected with anti-WDFY3 antibodies (n = 3 independent experiments with similar results). HC refers to heavy-chain. **f** Cre⁻ and Cre⁺ BMDMs were incubated with ACs for 1 h. Unbound ACs were washed away and BMDMs were collected for measurement of LC3-II by western blot (n = 5 biological replicates. The image shows the representative blot. * denotes non-specific band). **g** BMDMs were incubated with Hoechst-labeled ACs to allow efferocytosis. After removal of unbound ACs, BMDMs were collected and treated with digitonin to remove non-membrane bound LC3, and then immunostained for LC3 that is lipidated and membrane-bound. LC3-II staining was then quantified by flow cytometry for BMDMs that had engulfed Hoechst-labeled ACs (n = 5 biological replicates). Data are presented as mean ± SEM in (**b**, **c**, **f**), or as median ± 95% CI in (**a**, **g**). Two-sided P values were determined by a two-way ANOVA with Tukey's multiple comparisons test in (**b**, **c**, **f**), or by Mann–Whitney test in (**a**, **g**).

WDFY3 is downstream of PI3K-mediated F-actin disassembly, with PI3K inhibitor treatment, knockout of *Wdfy3* should not further impair AC uptake. In fact, with PI3K inhibition, *Wdfy3* knockout BMDMs showed lower AC uptake compared with control BMDMs (Supplementary Fig. 5f), supporting that WDFY3 affects AC uptake at least partly through PI3K and GAP-independent mechanisms. As expected, when PI3K is inhibited, uptake of 10 μm beads was comparable between *Wdfy3* knockout and control BMDMs (Supplementary Fig. 5g), suggesting that WDFY3-mediated regulatory mechanisms are not required for beads engulfment. Consistently, the percentage of BMDMs with F-actin surrounded beads was also comparable between *Wdfy3* knockout and control BMDMs (Supplementary Fig. 5h and Supplementary Movie 2), in sharp contrast to the higher percentage of F-actin ring surrounded engulfed ACs in *Wdfy3* knockout BMDMs compared with control BMDMs (Fig. 2h). Several studies have reported that macrophages more effectively engulf rigid cargos than soft cargos[29,30], likely because soft cargos deform and thus requiring stronger force generation[31]. We speculate that WDFY3-mediated F-actin dynamics is essential for the engulfment of the more challenging cargos, such as ACs, while dispensable for the uptake of the less challenging cargos, such as cargos with smaller size or higher rigidity. Our observations account for the differential requirement for WDFY3 during efferocytosis and pave the way for further interrogating the complex molecular mechanisms employed by macrophages in cargo-specific phagocytosis.

We have also validated the role of *Wdfy3* in macrophage efferocytosis in *Wdfy3*^fl/fl mice generated by the Knock-Out Mouse Project (KOMP) with two loxp sites flanking exon 8, and maintained on C57BL/6N background[32]. Breeding to *LysMCre* mice led to efficient knockout of *Wdfy3* though a small amount of residual protein remained detectable (Supplementary Fig. 6a). Consistently, we have observed impaired uptake of ACs in Cre^+ BMDMs (Supplementary Fig. 6b), further confirming that the role of *Wdfy3* knockout in macrophage efferocytosis is independent of the genetic strain or specific gene-inactivating mutation of the mouse models.

## WDFY3 deficiency led to impaired degradation of engulfed ACs

Sustained accumulation of periphagosomal F-actin prevents efficient phagosome-lysosome fusion[28]. We thus reasoned that defective actin depolymerization may impair the degradation of the engulfed cargos. To test the hypothesis, we determined the degradation of the engulfed ACs by Cre^+ and Cre^- BMDMs. We first incubated BMDMs with PKH26-labeled ACs for efferocytosis. After 60 min of incubation, unbound ACs were washed away and BMDMs were returned to the incubator for 3 h to allow degradation of the engulfed cargos. BMDMs were then fixed and imaged. We counted the percentage of AC^+ BMDMs that showed non-fragmented PKH26 staining implicating impaired degradation. Indeed, the percentage of BMDMs with non-fragmented ACs was greater in Cre^+ vs. Cre^- BMDMs (Fig. 3a), confirming impaired degradation in *Wdfy3*-deficient BMDMs (the quantification strategy for non-fragmented ACs is illustrated in Supplementary Note 3). To further validate the degradation defects using an alternative approach via flow cytometry-based quantification[26], we incubated BMDMs with ACs labeled by TAMRA, a dye that labels peptides and proteins, for efferocytosis and quantified the degradation of TAMRA signals 16 h post-efferocytosis. Consistently, *Wdfy3* knockout led to decreased rates of corpse degradation (Fig. 3b).

To dissect if the impaired degradation in *Wdfy3* knockout BMDMs is also linked to impaired lysosomal acidification, we dual-labeled ACs with Hoechst that stains DNA and is pH-insensitive, and pHrodo-Red that is pH-sensitive and shows fluorescent signals only under an acidified environment in the phagolysosome. *Wdfy3* knockout BMDMs showed a lower efferocytosis of Hoechst-labeled ACs (Fig. 3c, left panel), consistent with the results in Fig. 2c. For BMDMs with engulfed Hoechst^+ ACs, the percentage of pHrodo^+/Hoechst^+ BMDMs in Hoechst^+ BMDMs is lower in Cre^+ BMDMs vs. Cre^- BMDMs, supporting impaired acidification in *Wdfy3* knockout BMDMs (Fig. 3c, right panel). We observed consistent results using peritoneal macrophages (PMs) (Supplementary Fig. 7). Thus, WDFY3 is required for both the uptake and the degradation of engulfed ACs during efferocytosis, and *Wdfy3* knockout led to impaired acidification of the phagolysosome.

## WDFY3 deficiency led to defects in LAP

We next asked if the impaired degradation in *Wdfy3* knockout BMDMs was merely a consequence of the defects in actin depolymerization during phagosome formation or mediated by other potentially independent mechanisms. We first considered whether WDFY3 is involved in LC3-associated phagocytosis (LAP), a process by which LC3-II conjugation to phagosomes enables phagosome-lysosome fusion and AC corpse degradation[33–38]. Our hypothesis is built on the known role of WDFY3 in autophagic clearance of aggregated proteins, i.e. aggrephagy[39,40]. Specifically, The C-terminus of both mouse and human WDFY3 contains several functional domains (as illustrated in Fig. 3d)[41]. Co-immunoprecipitation and colocalization studies indicated that WDFY3 scaffolds a complex containing the p62-positive, ubiquitinated, aggregation-prone protein and the core autophagy proteins ATG5, ATG12, ATG16L1 and LC3/GABARAP[39,42]. The human ortholog of the yeast Atg8 includes the LC3 family (LC3A, LC3B, LC3B2 and LC3C) and the GABARAP family (GABARAP, GABARAPL1 and GABARAPL2). During aggrephagy, the WD40 repeats of WDFY3 are essential for its colocalization and interaction with ATG5[39]. The ATG5-ATG12 complex is required for an early stage of autophagosome formation, and together with the membrane-bound ATG16L1 facilitate the conjugation of LC3/GABARAP proteins to phosphatidylethanolamine, i.e. LC3 lipidation to form LC3-II, for autophagosome formation[43,44]. Recent work using HEK293T cells further revealed that WDFY3 has a conserved LIR (LC3-interacting region) motif in its WD40 region that directly binds to GABARAP, responsible for its recruitment to LC3B during aggrephagy[42].

We first set out to determine if endogenous WDFY3 interacts with GABARAP in macrophages. Whole-cell lysates from Cre^- and Cre^+ BMDMs were incubated with anti-GABARAP antibodies and were immunoprecipitated using protein A/G agarose beads. WDFY3 can be found in a complex with endogenous GABARAP in Cre^- BMDMs, confirming WDFY3 and GABARAP interactions (Fig. 3e). No precipitation was observed in *Wdfy3* knockout BMDMs, confirming the specificity of the antibody (Fig. 3e).

We thus reasoned that WDFY3 interacts with GABARAP, regulating the recruitment and lipidation of LC3 during LAP for subsequent cargo degradation. Consistent with previous literature[45], AC engulfment led to increased LC3-II as determined by western blot (Fig. 3f). The increase was blunted in *Wdfy3*-deficient BMDMs (Fig. 3f). We further confirmed the results using a flow cytometry-based assay[46]. Specifically, BMDMs were incubated with Hoechst-labeled ACs to allow efferocytosis. After removal of unengulfed ACs, BMDMs were collected and treated with digitonin to remove non-membrane bound LC3, and then immunostained for LC3 that is lipidated and membrane-bound, i.e. LC3-II. As quantified by flow cytometry, for BMDMs that had engulfed Hoechst-labeled ACs, *Wdfy3* knockout BMDMs had lower membrane-bound LC3-II (Fig. 3g), supporting impaired LC3 lipidation.

Taken together, WDFY3 regulates LAP-mediated degradation of engulfed ACs through interacting with GABARAP and facilitating LC3 lipidation and the subsequent phagolysosomal degradation.

## A C-terminus fragment of WDFY3 is sufficient for regulating degradation yet the full-length protein is required for the AC uptake during efferocytosis

It has previously been shown that a 1000 amino acid C-terminus fragment, that contains the PH-BEACH, WD40, LIR, and FYVE domains of WDFY3 or the D. melanogaster ortholog, Bluecheese, was sufficient

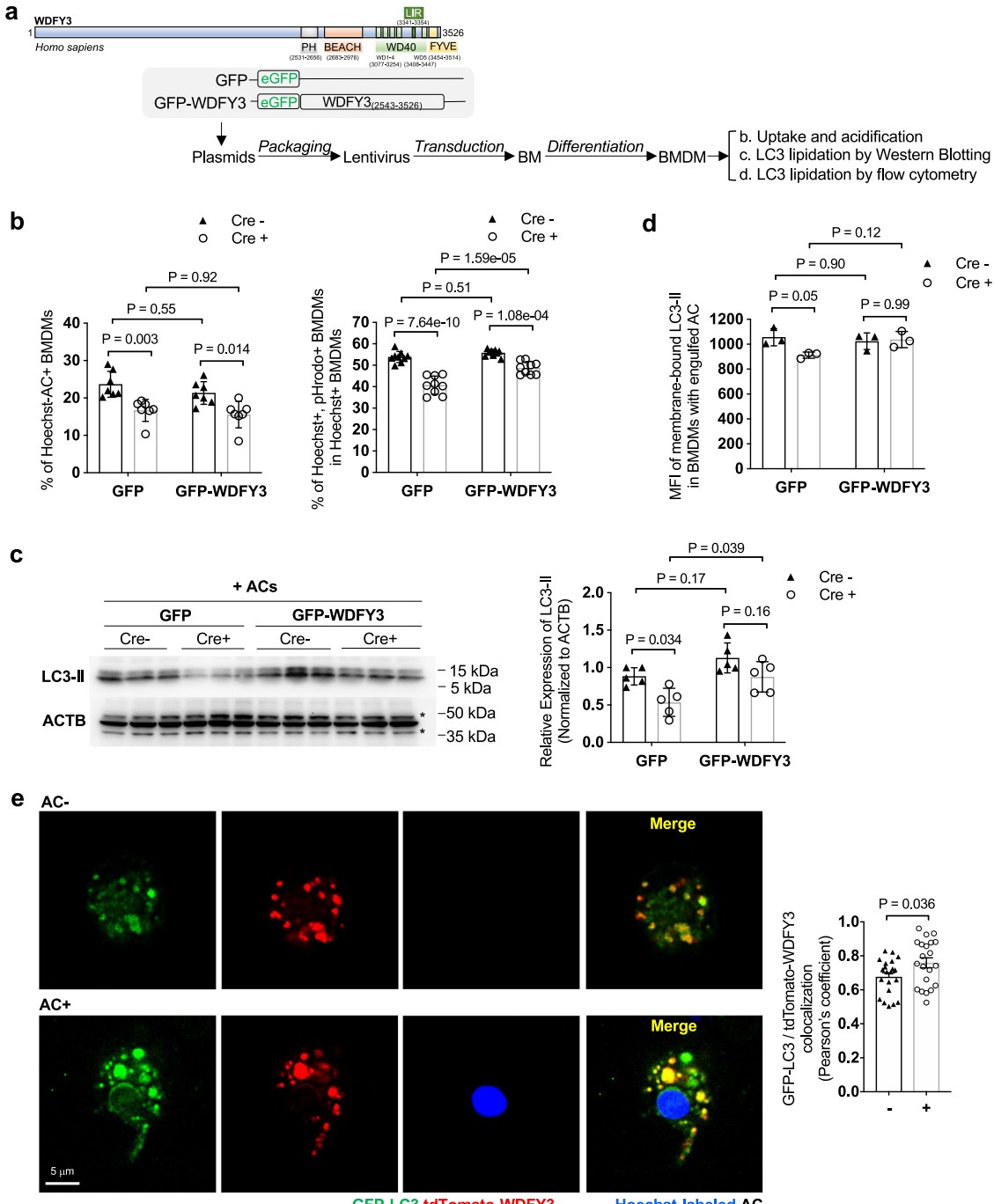

**Fig. 4 | The C-terminal WDFY3 is sufficient for regulating degradation yet the full-length WDFY3 is required for the uptake of ACs during efferocytosis.**
**a** Schematics of lentiviral overexpression of C-terminal WDFY3 in BMDMs of Cre⁻ and Cre⁺ mice. **b** C-terminal WDFY3 did not restore uptake, yet partially rescued the defects in cargo acidification in Cre⁺ mice (*n* = 7 biological replicates, each from the average of 2 technical replicates). **c** C-terminal WDFY3 restored LC3-II levels in Cre⁺ mice as determined by western blot (*n* = 5 biological replicates. * indicates non-specific bands) and in (**d**) by flow cytometry (*n* = 3 biological replicates). **e** BMDMs from GFP-LC3 mice were transfected with tdTomato-fused

C-terminal WDFY3(2981-3526) plasmid via electroporation. BMDMs were fed with Hoechst-labeled ACs. Unengulfed ACs were washed away and BMDMs were imaged to visualize GFP-LC3 phagosome association, C-WDFY3 intracellular localization, and GFP-LC3/tdTomato-WDFY3 colocalization with and without AC engulfment (*n* = 21 cells from 5 biological replicates. Images are representatives of 5 independent experiments each using one GPF-LC3 mouse). Data are presented as mean ± SEM. Two-sided *P* values were determined by a two-way ANOVA with Tukey's multiple comparisons test in (**b–d**) or by unpaired *t* test in (**e**).

to enhance the degradation of aggregated proteins in otherwise wild-type cells[39,47]. We therefore asked if this fragment was sufficient to regulate uptake and/or degradation during efferocytosis. We used lentiviral transduction to express C-terminal WDFY3 in both Cre⁻ and Cre⁺ BM cells that were then differentiated to BMDMs (as illustrated in Fig. 4a). Although expression of C-terminal WDFY3(2543-3526) did not

rescue the defective uptake in Cre⁺ BMDMs (Fig. 4b), it was sufficient to partly rescue the defects in the acidification of the engulfed ACs (Fig. 4b). Mechanistically, expression of C-terminal WDFY3 restored LC3 lipidation as quantified by both western blot (Fig. 4c) and flow cytometry (Fig. 4d). Thus, *Wdfy3* knockout led to two major defects affecting lysosomal acidification: (1) WDFY3 deficiency disrupted its

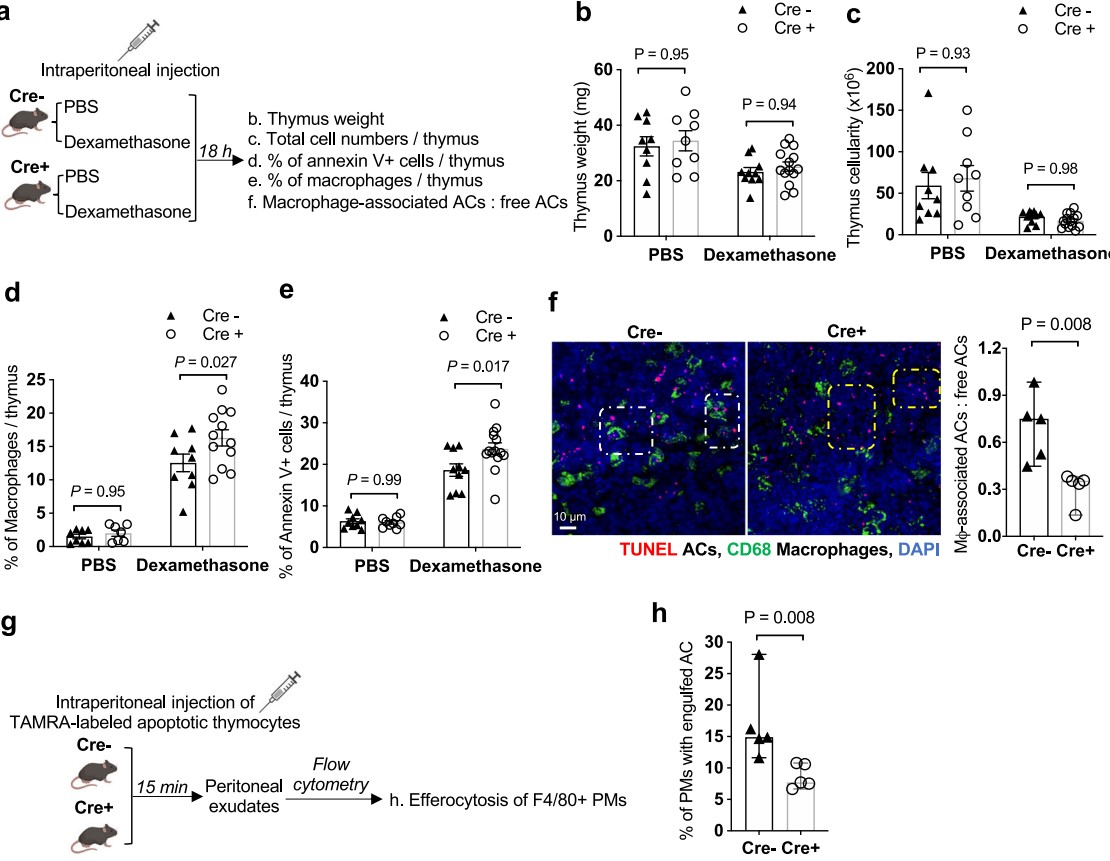

**Fig. 5 | Mice with myeloid *Wdfy3* knockout show impaired efferocytosis in vivo.**
**a** Schematics of experimental design for in vivo thymus efferocytosis assay.
**b** Thymus weight. **c** Total number of cells per thymus. **d** Percentage of F4/80[+]
macrophages in the thymus determined by flow cytometry. **e** Percentage of
Annexin V[+] ACs per thymus determined by flow cytometry. A higher percentage
implies impaired efferocytic clearance. **f** Thymic sections were stained with TUNEL
for ACs, and CD68 for macrophages. The ratio of macrophage-associated TUNEL[+]
cells vs. free TUNEL[+] cells was quantified and summarized. The white and yellow
squares highlight the macrophage-associated and free TUNEL[+] cells, respectively
($n = 5$ biological replicates). **g** Schematics of experimental design for in vivo

peritoneal macrophage efferocytosis assay. **h** Peritoneal exudates were stained for
F4/80 and the percentage of TAMRA[+] peritoneal macrophages was determined by
flow cytometry ($n = 5$ biological replicates). $n = 9$ Cre[-] and 9 Cre[+] biological repli-
cates for PBS group, $n = 10$ Cre[-] and 14 Cre[+] biological replicates for Dex-
amethasone group in (**b**, **c**, **e**). $n = 8$ Cre[-] and 7 Cre[+] biological replicates for PBS
group, $n = 9$ Cre[-] and 12 Cre[+] biological replicates for Dexamethasone group in (**d**).
Data are presented as mean ± SEM in (**b**–**e**), as median ± 95% CI in (**f**) and (**h**). Two-
sided *P* values were determined by a two-way ANOVA with Tukey's multiple com-
parisons test in (**b**–**e**), or by Mann–Whitney test in (**f**) and (**h**).

direct interaction with GABARAP/LC3 complex that facilitates LC3
lipidation and phagosome-lysosome fusion and subsequent acidifica-
tion; (2) Knockout of *Wdfy3* led to defects in F-actin disassembly, which
is expected to delay the subsequent phagosome-lysosome fusion and
lysosomal acidification. Therefore, though the C-WDFY3 completely
rescued LC3 lipidation, the acidification defects were partially rescued,
likely because C-WDFY3 was not sufficient to rescue defects in uptake.

Demonstrating WDFY3 localization and LC3 phagosome recruit-
ment using microscopy is critical to strengthen further the conclusion
on the role of WDFY in LAP. Because of the lack of a reliable antibody
for immunofluorescence staining of WDFY3 and the technical chal-
lenge to package the full-length *WDFY3* cDNA (which is 10.8 kb thus
preventing effective transfection or transduction), we fused tdTomato
to C-terminal WDFY3 and transfected the construct via electroporation
to BMDMs from *GFP-LC3* transgenic mice and imaged the cells with or
without AC engulfment. As shown in Fig. 4e, C-WDFY3 showed cyto-
plasmic localization. Without AC engulfment, BMDMs from *GFP-LC3*
mice showed basal levels of LC3 punta. With AC engulfment, LC3 punta
showed association with the phagosome. LC3 and C-WDFY3 colocali-
zation also increased in AC-engulfed BMDMs vs. BMDMs without AC
uptake. Thus, the imaging data are consistent with western blot and
flow cytometry data, supporting the role of WDFY3 in interacting with
LC3 complex and facilitating LC3 lipidation in LAP.

## Wdfy3 knockout subtly affects the transcriptome of BMDMs without affecting macrophage differentiation

To gain an unbiased view of how *Wdfy3* knockout affects the tran-
scriptomic signature of macrophages, we performed RNA-seq in Cre[-]
and Cre[+] BMDMs ($n = 4$ male mice, Supplementary Fig. 8). We first
confirmed that many receptors responsible for efferocytosis and
phagocytosis, including *Fcgr1*, *Fcgr2b*, *Fcgr3*, *Mertk*, *Timd4*, and many
macrophage marker genes, were expressed at similar levels between
Cre[-] and Cre[+] BMDMs (Supplementary Data 7). Using a FDR-adjusted *P*
value <0.05 and absolute fold-change > 1.5, only a small number of
genes were identified as differentially expressed (DE) between Cre[-] and
Cre[+] BMDMs, i.e., 20 genes were upregulated while 32 genes were
downregulated in Cre[+] vs. Cre[-] BMDMs (Supplementary Fig. 8a and
Supplementary Data 7).

We reasoned that modest changes in the expression of genes
belonging to the same pathway may imply functional impact. We thus
performed gene-set enrichment analysis (GSEA) to determine which
gene sets or pathways were enriched in upregulated or downregulated
genes due to *Wdfy3* knockout. The upregulated genes in Cre[+] BMDM
were enriched for Human Reactome Pathway terms, IL-4 and IL-13
Signaling and Collagen Formation, and GO Biological Process term,
Regulation of Chemotaxis (Supplementary Fig. 8b for representative
plots, and Supplementary Data 8 and Supplementary Data 9 for the

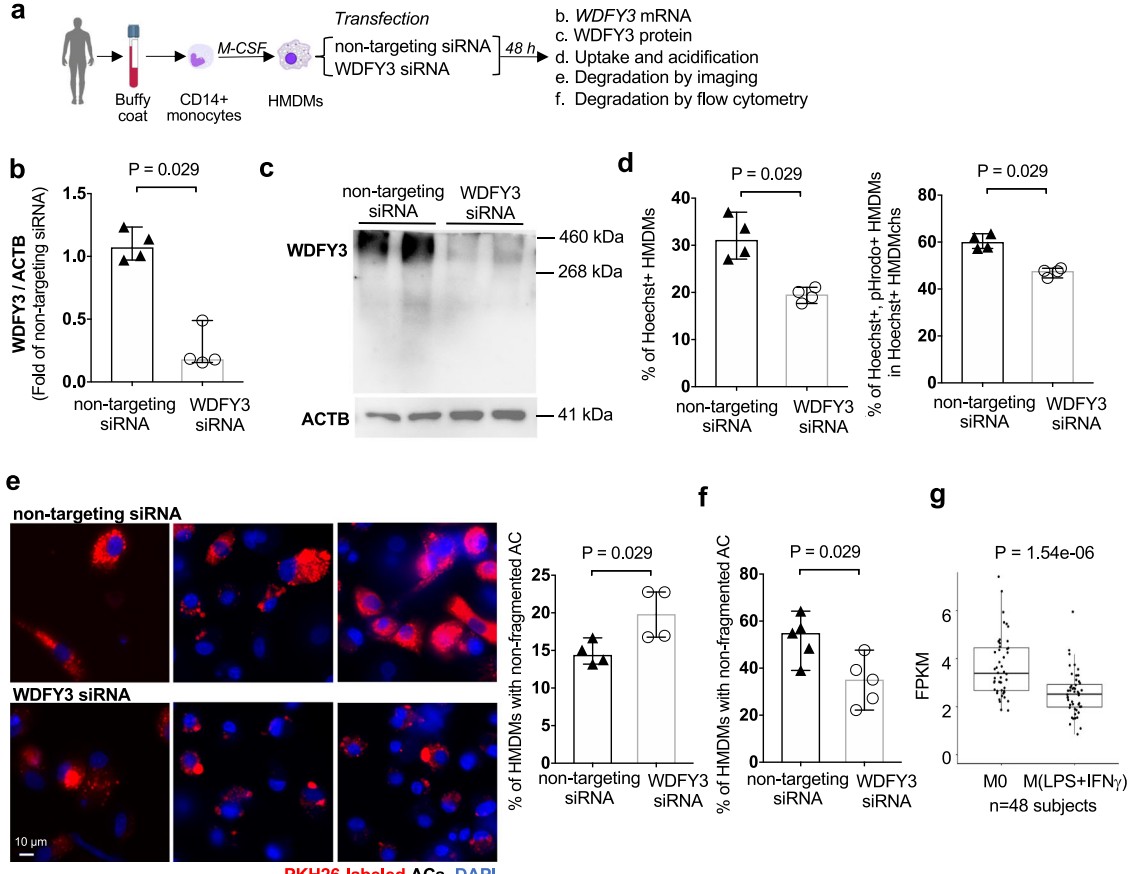

**Fig. 6 | WDFY3 regulates efferocytosis in human macrophages. a** Schematics of human monocyte differentiation to macrophages (HMDMs) and knockdown of *WDFY3* with Lipofectamine RNAiMAX-mediated transfection of siRNAs targeting *WDFY3*, or non-targeting siRNAs as the control. **b** Validation of knockdown efficiency at mRNA level by qRT-PCR ($n = 4$ independent experiments, each from the average of 3 technical replicates). **c** Validation of knockdown efficiency at protein level by western blot ($n = 2$ biological replicates, data are representative of 3 independent experiments). **d** Efferocytosis of apoptotic Jurkat cells labeled by both Hoechst and pHrodo. The percentage of HMDMs with Hoechst-labeled ACs (indicating uptake), and the percentage of Hoechst$^+$/pHrodo$^+$ HMDMs in Hoechst$^+$ HMDMs (indicating acidification upon uptake) were quantified by flow cytometry. Both uptake and acidification of ACs were impaired in HMDMs with siRNA-mediated *WDFY3* knockdown ($n = 4$ independent experiments, each from the average of 2 technical replicates). **e** Fragmentation of engulfed ACs was assessed 3 h after washing away the unengulfed ACs. The percentage of HMDMs with non-fragmented PKH26 staining in all PKH26$^+$ HMDMs was determined ($n = 4$ independent experiments). **f** Flow cytometry-based degradation assay was performed in HMDMs with procedures as described in Fig. 3b ($n = 5$ independent experiments). **g** RNA-seq was performed for HMDMs either unstimulated (M0) or treated with LPS and IFNγ for 18-20 h (M1-like). The expression of WDFY3 was visualized ($n = 48$ biological replicates). The box shows Q1, median, and Q3; the whiskers show 1.5 x interquartile range, though the lower whiskers in this plot only extend to the minimum as the minimum values are greater than the values corresponding to the lower whiskers. There is one outlier in M0 with FPKM 7.404. There are two outliers in M1-like with FPKMs 5.943 and 4.345. Data are presented as median ± 95% CI. Two-sided *P* values were determined by Mann–Whitney test.

complete GSEA output). The downregulated genes in Cre$^+$ BMDM were enriched for Human Reactome Pathway term, Peroxisomal Lipid Metabolism, and Gene Ontology (GO) Biological Process term, Fatty Acid Catabolic Process (Supplementary Fig. 8c for representative plots, and Supplementary Data 10 and Supplementary Data 11 for the complete GSEA output). Overall, no clear proinflammatory or anti-inflammatory gene signatures were identified in Cre$^+$ BMDMs.

We thus confirm that: (1) Despite the profound role of WDFY3 in AC uptake and degradation, the observed transcriptomic modifications by *Wdfy3* knockout were modest; (2) *Wdfy3* knockout did not affect macrophage maturation, as macrophage marker genes were not differentially expressed. We further confirmed that the percentage of F4/80$^+$ macrophages in BMDMs and PMs was comparable between Cre$^-$ and Cre$^+$ mice (Supplementary Fig. 9a). Population doubling during BMDM differentiation was not different between Cre$^-$ and Cre$^+$

mice, supporting comparable differentiation and proliferation capacity (Supplementary Fig. 9b).

## Mice with myeloid Wdfy3 knockout show impaired efferocytosis in vivo

To determine if *Wdfy3* knockout affects efferocytosis in vivo, we performed two in vivo efferocytosis assays in Cre$^-$ and Cre$^+$ mice as illustrated in Fig. 5a (thymus efferocytosis) and Fig. 5g (PM efferocytosis).

For thymus efferocytosis (Fig. 5a), we treated Cre$^-$ and Cre$^+$ mice with dexamethasone that induces apoptosis of thymocytes, using PBS as the control. 18 h after injection, thymi were isolated and weights were measured. The total number of cells per thymus was determined by dissociating one lobe of the thymus to count the cell number and then normalized to the weight of both lobes of the thymus. The dissociated cells were stained for Annexin V, a marker of apoptosis, and

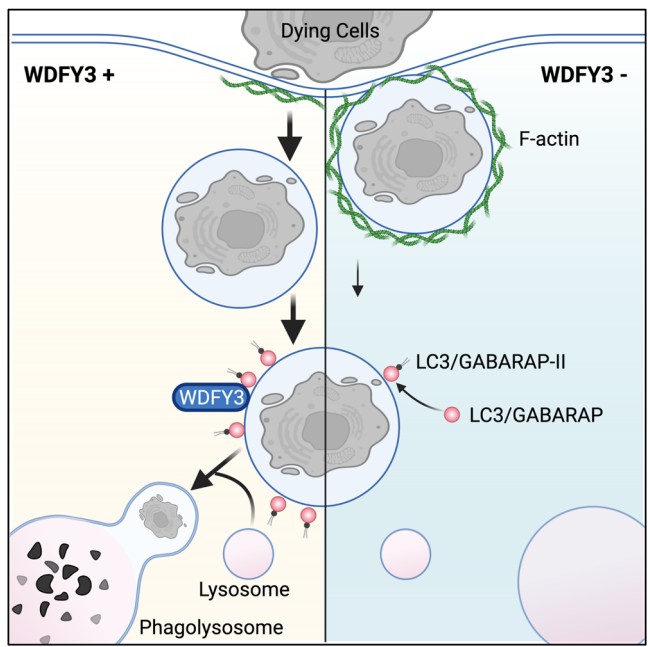

**Fig. 7 | Schematic figure summarizing how WDFY3 regulates macrophage efferocytosis.** WDFY3 is discovered as a new regulator of efferocytosis by macrophages. WDFY3 deficiency in macrophages specifically impaired uptake, not binding, of apoptotic cells due to defective actin depolymerization, thus phagosome formation. WDFY3 directly interacts with GABARAP, one of the seven members of the LC3/GABARAP protein family, to facilitate LC3 lipidation and the subsequent phagosome-lysosome fusion and degradation of the engulfed AC components.

macrophage marker F4/80, for quantification by flow cytometry (gating strategies are shown in Supplementary Fig. 10a). As expected, in dexamethasone-treated mice, coupled processes of thymocyte apoptosis and phagocytic clearance of dead cells led to reduced thymus weight (Fig. 5b) and the total number of cells per thymus (Fig. 5c), accompanied by increased macrophage infiltration (Fig. 5d) and a higher percentage of Annexin V$^+$ cells in the thymus (Fig. 5e). We did not observe a significant change in thymus weight or the total number of cells in Cre$^+$ mice compared to Cre$^-$ controls, yet myeloid *Wdfy3*-knockout led to an increased percentage of Annexin V$^+$ cells (Fig. 5e), implying impaired efferocytic clearance of apoptotic thymocytes (Fig. 5e). Note that the impaired efferocytic clearance in Cre$^+$ mice was unlikely to be caused by reduced macrophage availability because the percentage of macrophages per thymus was not lower in Cre$^+$ mice vs. Cre$^-$ mice treated with either PBS or dexamethasone (Fig. 5d). Induction of apoptosis by dexamethasone was equivalent between Cre$^-$ and Cre$^+$ mice for both thymocytes and BMDMs (Supplementary Fig. 11a and Supplementary Fig. 11b, respectively), ensuring that the increased percentage of Annexin V$^+$ cells in Cre$^+$ mice was due to impaired clearance, not altered induction of apoptosis, To assess efferocytosis in the thymus in situ, thymus sections were labeled and fluorescently imaged for TUNEL$^+$ cells (ACs) that were either associated with CD68$^+$ macrophages as a result of efferocytosis, or not associated with macrophages, i.e. free ACs, indicating inefficient efferocytic clearance. The ratio of macrophage-associated vs. free ACs was significantly lower in Cre$^+$ mice, further supporting impaired efferocytosis in *Wdfy3* knockout mice (Fig. 5f).

For PM efferocytosis (Fig. 5g), TAMRA-labeled apoptotic thymocytes were injected intraperitoneally into Cre$^-$ and Cre$^+$ mice. 15 min after injection, peritoneal exudate was collected and stained for F4/80 to identify macrophages. The percentage of TAMRA$^+$ PMs was quantified by flow cytometry (gating strategies are shown in Supplementary

Fig. 10b). Consistent with thymus efferocytosis, the percentage of TAMRA$^+$ PMs was significantly lower in Cre$^+$ mice (Fig. 5h), supporting reduced AC efferocytosis by PMs in vivo.

### WDFY3 is required for efferocytosis in human macrophages
We further confirmed that in human macrophages, knockdown of WDFY3 by transfection of small interfering RNA (siRNA) led to impaired uptake and degradation of engulfed ACs during efferocytosis (Fig. 6a–f). Human CD14$^+$ monocytes were isolated from buffy coats of three independent subjects and differentiated to macrophages (human monocyte-derived macrophages, HMDMs) using M-CSF. On day 5, non-targeting control siRNA pool or WDFY3-targeting siRNA pool were transfected using Lipofectamine RNAiMAX (Fig. 6a). At 48 h post-transfection, efficient knockdown of WDFY3 was confirmed at both mRNA (Fig. 6b) and protein levels (Fig. 6c). We then performed efferocytosis of human Jurkat cells labeled by both Hoechst and pHrodo. Consistent with the results in murine macrophages, both uptake and acidification (Fig. 6d) of ACs were impaired in HMDMs with siRNA-mediated WDFY3 knockdown. The percentage of HMDMs with non-fragmented ACs was greater with WDFY3 knockdown (Fig. 6e) (the quantification strategy for non-fragmented ACs is described in Supplementary Note 3). Flow cytometry analysis further confirmed lower degradation rate of TAMRA signals in HMDMs with WDFY3 knockdown, supporting impaired degradation of TAMRA-labeled ACs (Fig. 6f). We queried our previously published RNA-seq data for HMDMs either unstimulated (M0) or treated with LPS and IFNγ for 18–20 h (M1-like)[48], M1 stimulation reduced WDFY3 mRNA expression (Fig. 6g). Thus, reduced WDFY3 during inflammation and subsequently impaired WDFY3-mediated macrophage efferocytosis may contribute to impaired efferocytotic clearance of ACs in vitro and in vivo, exacerbating inflammation.

## Discussion
We developed a genome-wide CRISPR knockout screen in primary macrophages. By focusing on efferocytosis, a complex macrophage functional phenotype, we illustrated the versatility of pooled screens and provided an effective approach for genome-wide CRISPR screening in primary macrophages derived from Cas9 transgenic mice. We have identified many known genes regulating efferocytosis and general phagocytosis, illuminating the most important genes essential for the uptake of ACs during efferocytosis. We have also uncovered and validated WDFY3 as a novel regulator specifically regulating the phagocytosis of dying cells, but not other substrates, using orthogonal assays in vitro and in vivo. Mechanistically, WDFY3 deficiency led to impaired phagosome formation due to defects in actin depolymerization. We further revealed that WDFY3 directly interacts with GABARAP, one of the seven members of the LC3/GABARAP protein family, to facilitate LC3 lipidation and the efficient degradation of the engulfed cargo (Fig. 7 for a schematic summary). Further, WDFY3 expression was suppressed by inflammatory stimulation. Thus, WDFY3 regulates multiple steps during efferocytosis. Targeting WDFY3 may have therapeutic implications for diseases related to defective efferocytosis.

We unexpectedly uncovered a novel role of WDFY3 in LAP. The detailed molecular mechanisms require further investigation. The FYVE domain binds to phosphatidylinositol 3-monophosphate (PI3P). PI3P is produced during both autophagosome and phagosome formation and is required for the recruitment of autophagic machinery for downstream fusion with lysosomes. It is therefore plausible that during efferocytosis, WDFY3 is recruited through its known PI3P binding domain and acts as a scaffold that bridges ACs and autophagic machinery to regulate phagosome-lysosome fusion and lysosomal degradation of the engulfed cargos. Other questions remain unanswered, e.g. whether identical or different functional domains and binding partners have been involved in efferocytosis vs. in WDFY3-

mediated aggrephagy; what is the function of N-terminal WDFY3; which molecular domains are required and sufficient for the role of WDFY3 in uptake and/or degradation and what are their protein-protein interaction partners; what are the molecular mechanisms by which WDFY3 regulates F-actin disassembly and how WDFY3-mediated F-actin dynamics specifically regulate AC uptake. Pull-down experiments using specific domains of WDFY3 followed by quantitative proteomic screening, and live-cell imaging of endogenously tagged WDFY3 at baseline and during efferocytosis will further uncover these molecular mechanisms.

WDFY3 was highly expressed in myeloid cells compared with other immune cells. WDFY3 expression in HMDM was reduced by proinflammatory stimulation with LPS and IFNγ. The role of WDFY3-mediated efferocytosis in inflammation resolution and the therapeutic potential to enhance WDFY3 in diseases related to defective efferocytosis warrant further investigation. Indeed, overexpression of C-terminal WDFY3 (WDFY3$_{2981-3526}$) can enhance aggrephagy in neurons as indicated by increased aggregate clearance[39,47], supporting the therapeutic premise to target macrophage WDFY3 to stimulate efferocytosis. Therapeutic activation of WDFY3 may represent a pro-efferocytotic therapy in atherosclerosis and other diseases related to defective efferocytosis.

The screen has implied many highly ranked, potentially novel regulators of macrophage efferocytosis. Among the top-ranked positive regulators, in addition to *Wdfy3*, *Sh3glb1*, *Snx24*, and *Vps33a* are annotated in autophagy-related pathways (Supplementary Data 3). The generation of PI3P on the phagosomal membrane recruits LC3-conjugation machinery, and abrogation of LC3 lipidation at the membrane impairs phagosome maturation and lysosome-mediated degradation[1,34]. SH3GLB1 activates lipid kinase activity of PIK3C3 during autophagy by associating with the PI3K complex II (PI3KC3-C2)[49]. SNX24 contains a PX domain that mediates specific binding to membranes enriched in PI3P[50]. VPS33A is required for lysosome fusion with endosomes and autophagosomes[51]. These top screen hits may represent additional novel components of the cellular machinery that regulates efferocytosis. These promising targets and other potential novel regulators as uncovered by the screen have tremendous potential for additional novel discoveries. In our genome-wide screen, we employed a strategy of interpreting the results with a relatively permissive FDR threshold. Secondary screens with an increased number of gRNAs per gene and the number of cells infected per gRNA are expected to further improve the specificity and sensitivity for pooled screens in primary cells[52].

Furthermore, this screening platform can be adapted to screen for phagocytic regulators of distinct substrates, e.g. bacteria and amyloid-β aggregates, for which the engulfment by physiologically-relevant primary macrophages will be more informative, and to study gene pairs with epistatic interactions using libraries with multiplexed gRNAs. Our platform will facilitate the identification of efferocytosis regulators affecting distinct molecular steps, including recognition and degradation. For example, by applying different selection strategies to separate macrophages with engulfed and acidified cargos from those with engulfed yet non-acidified cargos, genes specifically regulating the intracellular processing and degradation can be systematically interrogated. Further, screening for regulators responsible for efferocytosis of dying cells undergoing different modes of cell death can be studied. Since the number of macrophages required for genome-wide coverage and the required Cas9 transgenic expression makes it impractical for genome-wide pooled screens to be performed in human primary macrophages, screens in primary murine macrophages provide opportunities for physiologically-relevant discoveries of novel biology, which can then be validated in human macrophages. Our experimental framework also provides a general strategy for systematic identification of genes of interest and uncovering novel regulators of complex macrophage functions, such as lipid uptake and foam cell formation. This genetic platform promises to accelerate clinically relevant, mechanism-based translational research projects in macrophage biology and related human diseases.

In summary, we have established a pooled genome-wide CRISPR knockout screen in primary macrophages for discoveries of novel regulators of macrophage efferocytosis. The screen has revealed WDFY3 as a regulator of efferocytosis in vitro and in vivo, in the mouse and in human cells. The findings advance our understanding of fundamental mechanisms of efferocytosis regulated by WDFY3. The screen top hits may likely contain additional novel regulators that can be further validated and promise to yield insights into diseases manifested by dysregulated efferocytosis. The innovative screen approaches established in this project are of broad and fundamental value to the community for conducting functional screens of novel regulators of complex macrophage function.

## Methods
The source of cell lines and primary cells (Supplementary Table 1), mouse strains (Supplementary Table 2), gRNA sequences (Supplementary Table 3), plasmids (Supplementary Table 4), primers for genotyping (Supplementary Table 5), primers for quantitative RT-PCR (Supplementary Table 6), siRNAs (Supplementary Table 7), antibodies (Supplementary Table 8), cell culture medium (Supplementary Table 9), chemicals and recombinant cytokines (Supplementary Table 10), assay kits (Supplementary Table 11), reagents for efferocytosis and phagocytosis assays (Supplementary Table 12), other reagents and supplies (Supplementary Table 13), and software (Supplementary Table 14) were summarized in the Supplementary Information.

### Cell lines
Cell lines, including Jurkat (lymphocytes, human acute T cell leukemia), THP-1 (monocytes, human acute monocytic leukemia), U937 (monocytes, human histiocytic lymphoma), and L-929 (mouse fibroblasts) were obtained from ATCC and handled according to the instructions provided on the ATCC product sheet. (Supplementary Table 1)

### Bone-marrow isolation and differentiation to bone marrow-derived macrophages (BMDMs)
Bone-marrow (BM) cells from 8 to 12 weeks old mice were isolated by flushing femurs and tibia with DMEM basal medium using 10 mL syringes with 22 G needles (day 0). The isolated BM cells were cultured at 37 °C, 5% $CO_2$ on non-tissue-culture-treated vessels for 7–10 days in BMDM culture medium containing DMEM supplemented with 10% (vol/vol) heat-inactivated fetal bovine serum (HI-FBS), 20% (vol/vol) L-929 fibroblast conditioned medium, and 2 mM L-Glutamine. During differentiation, the growth medium was replaced with fresh medium 96 h after seeding and then every 2–3 days. In vitro assays were performed in BMDMs from day 7 to day 10.

### Peritoneal macrophage (PM) isolation
PM isolation buffer (DPBS supplemented with 10% (vol/vol) HI-FBS and 2 mM EDTA) was injected into the peritoneum of 8–12 weeks old mice for a 10 min incubation. Peritoneal exudates were then collected using 10 mL syringes with 25 G needles and plated on non-tissue-culture-treated vessels. Unattached cells were removed 6 h after plating and the attached cells were used. PMs were maintained in DMEM supplemented with 10% (vol/vol) HI-FBS, 20% (vol/vol) L-929 fibroblast conditioned medium, and 2 mM L-Glutamine for 12–18 h at 37 °C, 5% $CO_2$ before the indicated assays[27].

### Human-monocyte-derived macrophages (HMDMs)
Buffy coats of anonymous, de-identified healthy adult volunteer donors were purchased from the New York Blood Center (NYBC), with

informed consent obtained by the NYBC, for isolation of peripheral blood mononuclear cells (PBMCs). Buffy coats were diluted with 1X DPBS supplemented with 2 mM EDTA at a 1:1 ratio, i.e. 8 mL buffy coats were diluted with 8 mL DPBS to a total volume of 16 mL. The diluted buffy coats were carefully laid on 9 mL Ficoll-Paque solution, i.e. a 4:3 ratio in 50-mL conical tubes and centrifuged at $400g$ for 40 min at 20 °C without brake. PBMC layer was transferred and washed with washing buffer (1X DPBS, 2% (vol/vol) HI-FBS, 5 mM EDTA, 20 mM HEPES and 1 mM sodium pyruvate), centrifuged at $500g$ for 10 min at 4 °C. The PBMC pellets were washed again in RPMI-1640 medium containing 20% (vol/vol) HI-FBS. The pellets were then resuspended and cultured in RPMI-1640 medium supplemented with 20% (vol/vol) HI-FBS and 50 ng/mL human macrophage colony-stimulating factor (M-CSF) for 7–10 days[53]. The growth medium was replaced with fresh medium 96 h after seeding and then every 2–3 days.

## THP-1 and U937 differentiation to macrophages

THP-1 human acute monocytic leukemia cell line was obtained from ATCC and grown in suspension in THP-1 culture medium containing RPMI-1640 supplemented with 10% (vol/vol) HI-FBS, 1 mM Sodium Pyruvate, 10 mM HEPES, and 50 μM 2-Mercaptoethanol. THP-1 macrophages were differentiated from THP-1 cells in the above culture media supplemented with 100 nM Phorbol 12-myristate 13-acetate (PMA) for 24 h at a seeding density of $1 \times 10^6$ cells per well of a 6-well tissue culture plate. PMA-containing media was then removed and replaced with THP-1 culture media for 48 h culture. The same seeding density was used for U937 differentiation to macrophages with 50 nM PMA for 3 days[11].

## Experimental animals

Animal protocols were approved by the Institutional Animal Care And Use Committee at Columbia University (Protocol Number AC-AABN5558). All animals were cared for according to the NIH guidelines. Mice were socially housed in standard cages at 22 °C with 40-60% humidity under a 12-12 h light-dark cycle with ad libitum access to water and food provided by the mouse barrier facility (PicoLab Rodent Diet 20 5053 and 5058, LabDiet). *Rosa26-Cas9* knockin mice were obtained from the Jackson Laboratory (Cat# 026179, C57BL/6J) (female mice were used for the CRISPR screen and validation). *Wdfy3$^{fl/fl}$* mice were obtained from Dr. Ai Yamamoto's lab (*Wdfy3$^{fl/fl}$*: 129/SvEv x C57BL/6 flanking Exon 6)[24] and Dr. Konstantinos Zarbalis's lab (*Wdfy3$^{fl/fl}$*: C57BL/6NJ flanking Exon 8)[32]. Myeloid-specific Wdfy3 knockout mice were created by crossing *LysMCre$^{+/-}$* mice (the Jackson Laboratory, Cat# 004781, C57BL/6 J) with *Wdfy3$^{fl/fl}$* mice. *LysMCre$^{+/-}$Wdfy3$^{fl/fl}$* mice (Cre$^+$) had myeloid-specific knockout of *Wdfy3*, while *LysMCre$^{-/-}$Wdfy3$^{fl/fl}$* littermates (Cre$^-$) served as controls. The *GFP-LC3* mice[54,55] were maintained as homozygote[47] (Background strain C57BL/6 J). The wild-type mice for thymocyte isolation were obtained from the Jackson Laboratory (Cat# 000664, C57BL/6J). Mice were euthanized by $CO_2$ asphyxiation followed by cervical dislocation. Both male and female mice were used at 8-12 weeks old unless otherwise specified, and experimental and control mice were co-housed. (Supplementary Table 2 for mouse strains)

## Lentiviral plasmid construction

The Brie murine CRISPR knockout pooled library in the lentiGuide-Puro backbone was obtained from Addgene (#73663)[17]. To validate the top screen hits using individual gRNAs, pairs of oligonucleotides with BsmBI-compatible overhangs were separately annealed and cloned into the lentiGuide-Puro vector (Addgene #52963) using standard protocols available via https://www.addgene.org/52963/. To validate the role of *Wdfy3* using a separate plasmid platform, gRNA targeting *Wdfy3* was selected from the murine Sanger lentiviral CRISPR library (Sigma) and the *Wdfy3*-targeting lentiviral vector, as well as the non-targeting control vector, were obtained (Sigma). To overexpress

C-terminal WDFY3, pLE4-eGFP-WDFY3$_{2543-3526}$ was constructed by inserting Myc-WDFY3$_{(2543-3526)}$, which was from pcDNA-myc-WDFY3$_{2543-3526}$ provided by Dr. Ai Yamamoto[39], into the pLE4 lentiviral backbone[56]. eGFP was then inserted into the N-terminal of WDFY3 to generate pLE4-eGFP-WDFY3$_{(2543-3526)}$ to allow the identification of WDFY3-overexpressing population by flow cytometry upon transduction.

## Lentiviral packaging and transduction

Lentivirus particles were generated from HEK293T cells (ATCC CRL-3216) by co-transfection of lentiviral vectors with the packaging plasmid psPAX2 (Addgene #12260) and envelope plasmid pMD2G (Addgene #12259) using FuGene 6 transfection reagent (Promega). The medium was changed 16-18 h after transfection. 24 h after media change, lentiviral supernatants were harvested and stored at 4 °C. Fresh media were fed and lentiviral supernatants were collected again 24 h later and pooled together with the first harvest. The pooled supernatants were then filtered through 0.45-μm SFCA filters (Corning). Lentiviral particles were further concentrated using Lenti-X concentrator (Takara Bio) following the manufacturer's instructions.

Mouse BM cells were isolated and plated (day 0). On day 1, BM cells were virally transduced in BMDM culture medium supplemented with 10 μg/mL polybrene. On day 2, half of the medium was replenished with fresh BMDM culture medium. On day 6, the transduced cells underwent puromycin selection at 5 μg/mL for 48 h. On day 9, i.e. 24 h after removing puromycin, BMDMs were used for efferocytosis assays. The pLE4 lentiviral vector does not have a puromycin-resistant gene, thus no antibiotics selection was performed. For pLE4 lentivector expressing GFP only or GFP-WDFY3, transduction was performed on day 0 and assays were performed on day 8.

## Induction of apoptosis and fluorescent labeling of apoptotic cells (ACs)

Apoptotic Jurkat cells were generated by treating Jurkat cells with 5 μg/mL staurosporine in RPMI-1640 medium for 3 h at a density of $2.5 \times 10^6$ cells/mL at 37 °C, 5% $CO_2$. The method routinely yields greater than 90% Annexin V$^+$ apoptotic Jurkat cells. After washing in 1X DPBS, apoptotic Jurkat cells were resuspended at a concentration of $2 \times 10^7$ cells/mL in Diluent C with either PKH67 (green fluorescence) or PKH26 (red fluorescence) per the manufacturer's instruction. After labeling, the cells were rinsed twice with DMEM basal medium containing 10% HI-FBS and immediately used for efferocytosis assay. For labeling with other fluorescent probes, ACs were resuspended at a density of $2.5 \times 10^6$ cells/mL in DMEM basal media and incubated with 20 ng/mL pHrodo red (Life Technologies) and/or Hoechst 33342 solution (20 mM, 1:10,000 dilution, Thermo Scientific) for 30 min, or NuclearMask Blue Stain solution (1:2000 dilution, Invitrogen) for 30 min. TAMRA staining was applied to ACs at a concentration of $2 \times 10^7$ cells/mL in DMEM basal medium at 10 μg/mL for 25 min.

To isolate mouse thymocytes and induce apoptosis, thymi were dissected from C57BL/6 J mice (~6-weeks) and were grounded and filtered through 70 μm cell strainer to obtain single-cell suspension. The induction of apoptosis can be initiated by one of the two approaches: (1) Incubating the thymocytes with 50 μM dexamethasone in DMEM at 37 °C, 5% $CO_2$ for 4 h; (2) UV irradiation (Analytik Jena UVP EL Series Lamps, UVP95020001) for a total of 12 min and then incubated for 2.5 h at 37 °C with 5% $CO_2$[25] and the percentage of Annexin V$^+$ cells that are apoptotic were confirmed to be >90%. Labeling of apoptotic thymocytes was performed as described above for Jurkat ACs.

## Preparation of sheep red blood cells (RBCs) for efferocytosis

Sheep red blood cells (RBCs) (Rockland) were obtained. For heat-shock treatment, RBCs were incubated under 56 °C in a water bath for 5 min[57]. For IgG-opsonization, RBCs were incubated with 1 μg/mL anti-

RBCs antibodies in DMEM basal medium containing 10% (vol/vol) HI-FBS to conjugate with IgG at 37 °C, 5% $CO_2$ for 1.5 h[57]. The non-treated, heat-shock treated or IgG-conjugated RBCs were labeled with PKH67 following the same procedures for the labeling of apoptotic Jurkat cells.

### In vitro efferocytosis and phagocytosis assays

For imaging-based quantification, macrophages were plated in 96-well plates at a density of $0.3 \times 10^5$ per well. For flow cytometry-based quantification, macrophages (BMDMs, PMs, or HMDMs) were plated in 6-well or 24-well plates at a density of $1.5 \times 10^6$ per well or $0.2 \times 10^6$ per well, respectively. Fluorescently-labeled apoptotic cells were co-incubated with macrophages at a 5:1 AC: macrophage ratio for 1 h (or as described in Figures) at 37 °C, 5% $CO_2$. Macrophages were then washed with 1X DPBS gently to remove unbound targets. For imaging-based quantification, macrophages were fixed with 2% PFA for 30 min, rinsed 3 times with 1X DPBS, and counterstained by DAPI. For flow cytometry-based quantification, macrophages were lifted using Cell-Stripper, a non-enzymatic cell dissociation solution, for live-cell analysis. The phagocytosis of beads, RBCs, and zymosan particles by BMDMs was determined upon incubation for 1 h at the specific ratio or concentration as specified in the respective figures.

To determine how inhibiting PI3K affects macrophage efferocytosis and phagocytosis, BMDMs were pretreated with 10 μM PI3K inhibitor LY294002 for 60 min and during efferocytosis. The percentage of AC-engulfed or beads-engulfed BMDMs was quantified by flow cytometry.

### CRISPR-Cas9 screen for efferocytosis in BMDMs and validation

CRISPR-Cas9 screens were performed using the Brie library[17]. BM cells isolated from *Rosa-Cas9* knockin mice were virally transduced at a low multiplicity of infection (MOI) of 0.3 and targeting ~1000 fold coverage of the library. After puromycin selection, BMDMs were dissociated and replated in 10-cm tissue culture plates at a density of $6 \times 10^6$ per plate for two-round efferocytosis. For the 1st round, PKH67-labeled ACs were incubated with BMDMs at a 5:1 ratio for 45 min. After removing the unbound ACs, macrophages were rested for 3 h before the 2nd round, in which PKH26-labeled ACs were incubated with BMDMs at a 5:1 ratio for 90 min. Unbound ACs were removed and BMDMs were collected for sorting on BD Influx. The sorted populations were processed individually for genomic DNA extraction using DNeasy Blood and Tissue Kit (Qiagen) and subjected to PCR reactions to amplified the gRNA sequences and generate the libraries. The purified PCR products were sequenced on Illumina NextSeq 500 system to determine gRNA abundance in two independent replicates. The fastq files were processed using count_spacers.py to obtain the gRNA counts[58] (refer to the Code Availability statement for code). The gRNA count matrix files were then analyzed using MAGeCK (Model-based Analysis of Genome-wide CRISPR-Cas9 Knockout[21], version 0.5.7). MAGeCK (mageck test) uses Robust Rank Aggregation (RRA) for robust identification of CRISPR-screen hits, and outputs the summary results at both sgRNA and gene level as ranked lists of screen hits. Independent validation of top screen hits by individual gRNAs was performed by lentiviral transduction of gRNA in *Rosa-Cas9* knock-in BM cells and differentiation to BMDM followed by efferocytosis assays and quantification[23].

Initial validation of top screening hits as shown in Fig. 1g and Supplementary Fig. 1 was performed using Nikon Ti-S Automated Inverted Microscope with NIS-Elements High Content Analysis Imaging Software according to the manual. Briefly, nuclei were segmented as primary objects by DAPI images. Cut-off size was optimized to remove improperly segmented cells, such as large debris and apoptotic bodies, from further analyses. Individual cell outlines were obtained by growing all initial nuclei regions simultaneously until they touch or reach the image border, i.e. Watershed. ACs were segmented by PKH26 images. Cut-offs to the size, roundness, and intensity of the signals were optimized to remove auto-fluorescent bodies and cell debris. The count feature was then used to count cells and cells with ACs, and calculate the percentage of cells with ACs. The analysis template was provided in the Github repository, which can be used as guidance rather than prescriptive, as differences in staining intensity can affect the effectiveness of segmentation.

### Analysis of macrophage capability of binding

BMDMs were stained with 0.5 μM CellTracker Green CMFDA (5-chloromethylfluorescein diacetate) for 60 min. The CellTracker dye freely passes through cell membranes and is well-retained in cells, allowing labeling of cytoplasmic area. BMDMs were then treated with 5 μM cytochalasin D for 30 min. Cytochalasin D blocks the assembly and disassembly of actin monomers, thus preventing internalization of ACs. The treated BMDMs were then incubated with TAMRA-stained apoptotic mouse thymocytes for 30 min at a 5:1 ratio of AC: BMDM at 37 °C, 5% $CO_2$ to allow binding. The unbound ACs were extensively washed with 1X DPBS, BMDMs were fixed with 2% PFA for 30 min and washed with 1X DPBS for 3 times, followed by imaging with ImageX-press Micro 4 High-Content Imaging System with a Nikon Plan Apo λ 20x/0.75 objective lens to analyze the percentage of macrophages with bound ACs. Imaging quantification is described in the Supplementary Note 1.

### Time-lapse imaging of phagosome formation

BMDMs cultured on chambered coverslips with 8 individual wells (ibidi) at a density of $0.12 \times 10^6$ were stained with 0.5 μM CellTracker Green CMFDA Dye (Invitrogen) in DMEM supplemented with 10% (vol/vol) HI-FBS for 60 min. The medium was replaced with fresh DMEM containing 10% HI-FBS and apoptotic Jurkat cells were added at a 5:1 AC: BMDM ratio. BMDMs were imaged with Nikon Ti Eclipse inverted microscope for spinning-disk confocal microscopy equipped with a 60x/1.49 Apo TIRF oil immersion lens. Images of the same fields were recorded at 30 s intervals for 20 min.

### Visualization and quantification of F-actin dynamics during efferocytosis of ACs or phagocytosis of beads

BMDMs plated on 96-well plates were stained with 0.5 μM CellTracker Green CMFDA Dye (Invitrogen) and 1 μM SiR-actin (Cytoskeleton) for 60 min. ACs labeled by NCS-nucleomask blue (Invitrogen) were added to the macrophages to replace the staining medium at a 5:1 AC: macrophage ratio for 1 h efferocytosis. Macrophage monolayer was then washed with 1X DPBS to remove unbound ACs, fixed with 2% PFA for 30 min and washed with 1X DPBS for 3 times, and imaged by ImageXpress Micro4 high-content microscopy (Molecular Device) with a Nikon Plan Apo λ 40X/0.95 objective lens. The percentage of macrophage with bright F-actin ring, as an indicator of F-actin polymerization, was quantified. Imaging quantification is described in the Supplementary Note 2.

To quantify F-actin intensity by flow cytometry, BMDMs plated on 6-well non-tissue culture plates were incubated with Hoechst-labeled ACs for 1 h. Unbound ACs were washed away and BMDMs were collected and fixed by 2% PFA for staining with 1 μM siR-actin in washing buffer (1X DPBS, 2% (vol/vol) HI-FBS, 5 mM EDTA, 20 mM HEPES and 1 mM sodium pyruvate). siR-actin-labeled F-actin levels were quantified as the mean fluorescence intensity (MFI) of siR-actin in BMDMs with or without engulfment of ACs.

To assess F-actin dynamics during phagocytosis of beads, BMDMs were stained with 1 μM CellMask-Green Actin Tracking Dye (Invitrogen) for 30 min. 10 μm red fluorescent polystyrene beads (Invitrogen) were added at a 5:1 ratio to macrophage for 1 h efferocytosis. The same procedures were followed as above to image and quantify the percentage of macrophage with F-actin rings. Scoring of F-actin ring was illustrated in Supplementary Movie 2.

## Analysis of fragmentation of engulfed AC components by imaging

PKH26-labeled ACs were added to BMDMs or HMDMs and incubated for 45 min. Unengulfed ACs were removed by vigorous rinsing with 1X DPBS. After being cultured for an additional 3 h, the macrophages were fixed with 2% PFA and counterstained with DAPI. Images were captured using ImageXpress Micro4 high-content microscopy (Molecular Device) with a Nikon Plan Apo λ 40X/0.95 objective lens. The percentage of macrophages containing non-fragmented AC-derived fluorescence, which is a measure of AC corpse degradation, was quantified[25]. Imaging quantification is described in Supplementary Note 3 for BMDMs and HMDMs.

## Analysis of degradation of engulfed AC by flow cytometry

TAMRA-labeled ACs were added to CellTracker Green stained BMDMs or HMDMs and incubated for 45 min. Unengulfed ACs were removed by 1X DPBS wash. Macrophages were harvested for flow cytometry to quantify the MFI of TAMRA⁺CellTracker⁺ population at baseline and 16 h after efferocytosis ended. The rate of degradation was calculated as (MFI of TAMRA at 0 h−MFI of TAMRA at 16 h)/MFI of TAMRA at baseline X 100%.

## Membrane-bound LC3 detection by flow cytometry

BMDMs were incubated with Hoechst-labeled ACs at a 5:1 AC: BMDM ratio at 37 °C, 5% $CO_2$ for 1 h efferocytosis. Unbounded ACs were washed away. BMDMs were collected and resuspended in 300 μL cold DPBS with 20 μg/mL digitonin, and incubated on ice for 10 min to permeabilize cells and allow non-membrane bound LC3 to be removed from cells. Permeabilized BMDMs were then centrifuged for 5 min at 750g, followed by incubation with anti-LC3A/B-FITC antibody or anti-LC3A/B-PE antibody diluted in cold washing buffer (1X DPBS, 2% (vol/vol) HI-FBS, 5 mM EDTA, 20 mM HEPES and 1 mM sodium pyruvate) for 15 min on ice to stain the membrane-bound lipidated LC3-II within the cells. After staining, macrophages were washed with 1 mL cold washing buffer and were centrifuged for 5 min at 750g. Cell pellets were resuspended in washing buffer and acquired on a flow cytometer[46].

## Membrane-bound LC3 detection by immunoblotting

UV-induced apoptotic thymocytes were added to BMDMs cultured in 6-well plate at a ratio of 5:1. After incubating for 1.5 h, BMDMs were washed 3 times with DPBS and harvested with CellStripper. Cells from each well were lysed with 70 μL RIPA lysis buffer (Millipore) supplemented with protease inhibitor cocktail (Roche) and phosphatase inhibitor cocktail (Roche) for 30 min on ice. Lysates were then centrifuged at 12,000g for 10 min and supernatant were transferred to a fresh tube. Protein concentration was quantified using Pierce BCA protein assay kit (Thermo Fisher). Equal amount of protein were mixed with 4X Bolt LDS sample buffer and 10X reducing reagent dithiothreitol (Novex Bolt Sample Reducing Agent, 10X). Samples were heated for 10 min at 60 °C and centrifuge at 12,000g for 30 s before loading to a 16% Tris-glycine gel. Proteins were then electro-transferred to a 0.45 μm (or 0.2 μm) PVDF membrane (Thermo Scientific). After blocking with 5% milk, the membrane was incubated with rabbit anti-LC3B primary antibody (ab48394, Abcam) overnight at 4 °C. The membrane was then washed for 3 times in TBST and incubated with HRP-conjugated goat anti-rabbit IgG (1:5000 dilution) for 1 h at room temperature. After the final wash to remove unbound antibodies, the protein expression was detected by SuperSignal™ West Pico PLUS Chemiluminescent Substrate (Thermo Scientific) and imaged using ChemiDoc Imaging System (Bio-rad). Band intensity was quantified using the software ImageJ.

## Immunoblotting of WDFY3

Macrophages cultured on 6-well plate were harvested and cells from one well were lysed in 70 μL RIPA lysis buffer (Millipore) supplemented with protease inhibitor cocktail (Roche) and phosphatase inhibitor cocktail (Roche). Protein concentration was quantified using Pierce BCA protein assay kit (Thermo Fisher). Equal amount of protein were mixed with 5X SDS sample buffer [5% (vol/vol) β-Mercaptoethanol, 0.02%(vol/vol) Bromophenol blue, 30% (vol/vol) Glycerol, 10%(vol/vol) Sodium dodecyl sulfate, 250 mM Tris-Cl, pH 6.8)] and loaded onto a 3–8% Tris-Acetate NuPage gel and then electro-transferred to a 0.45 μm (or 0.2 μm) PVDF membrane (Thermo Scientific). After blocking with 5% milk, the membrane was incubated with rabbit anti-WDFY3 primary antibody (Ai Yamamoto lab[40]) overnight at 4 °C. The membrane was then washed for 3 times in TBST and incubated with HRP-conjugated goat anti-rabbit IgG (1:5000 dilution) for 1 h at room temperature. After the final wash to remove unbound antibodies, the protein expression was detected by SuperSignal™ West Pico PLUS Chemiluminescent Substrate (Thermo Scientific) and imaged using ChemiDoc Imaging System (Bio-rad). The membranes were then blocked with 5% milk for 30 min followed by incubating with HRP-conjugated antibody to blot ACTB (β-actin) for 1 h at room temperature. After washing with TBST for 3 times, the membranes were imaged using ChemiDoc Imaging System (Bio-rad). Band intensity was quantified using the software image J.

## GABARAP immunoprecipitation

Around $20 \times 10^6$ BMDMs were harvested from two 10-cm dishes and lysed in 600 μL RIPA lysis buffer (Millipore) supplemented with protease inhibitor cocktail (Roche) on ice for 30 min. The lysates were centrifuge at 12,000g for 10 min. 450 μL supernatant was taken out and transferred into a fresh pre-chilled tube, followed by measuring the protein concentration with Pierce BCA protein assay kit (Thermo Fisher). For input sample preparation, about 5% of total protein was aliquoted to a separate tube and mixed with 4X Bolt LDS sample buffer as well as 10x reducing reagent for GABARAP blot. About 5% of total protein were aliquoted and mixed with 5X SDS sample buffer for WDFY3 blot. 10 μg of the input samples were loading to the gel.

For Pull-down sample preparation, 250 μg total cell lysates were incubated with 8 μg anti-GABARAP antibodies (Abcam, ab191888) in 500 μL RIPA buffer overnight at 4 °C. 100 μL protein A/G agarose beads (Thermo Scientific Pierce) were centrifuged at 600g and washed with RIPA buffer for 3 times and were then added to the antibody and lysates mixture for another 1–2 h at 4 °C. After incubation, the mixture were centrifuged at 600g for 1 min. After removing supernatant, beads were washed with 1 mL RIPA buffer for 3 times. Each time remained 100 μL at bottom. For eluting the immunoprecipitants from the antibody and beads, the sample after the final wash were separated to two tubes. One tube containing 40 μL beads was then mixed with 2X Bolt LDS sample buffer with 2x reducing reagent for GABARAP blotting. The remaining 40 μL beads were incubates with 2X SDS sample buffer for WDFY3 blotting. Samples were heated at 60 °C for 10 min and centrifuge at 800g to elute the proteins. Fifteen microliters of the pull-down samples were then subjected to 3–8% Tris-glycine gel for immunoblotting analysis.

## Live-cell imaging of GFP-LC3, tdTomato-WDFY3, and quantification of colocalization

BM cells were isolated from GFP-LC3 transgenic mice and then differentiated to BMDMs. At day 5, BMDMs were lifted up and transfected with pDEST-tdTomato-WDFY3₍₂₉₈₁₋₃₅₂₆₎[39], a plasmid construct with tdTomato-fused to the N-terminal of the C-terminal WDFY3₍₂₉₈₁₋₃₅₂₆₎, via electroporation using P3 primary cells kit (Lonza, V4XP-3032). After electroporation, cells were seeded into chambered coverslip (ibidi, 80826) to continue differentiation. At day 8 or day 9, apoptotic thymocytes were labeled with Hoechst for 30 min at 1:10,000, and fed to BMDMs for 15 min and washed with DPBS for 8–10 times. Note that a shorter incubation time and extensive washing step were used to ensure minimal amount of unengulfed ACs remaining on the coverslip

in order to perform live-cell imaging using a Nikon spinning-disk confocal microscope with Plan Apo λ 100X/0.95 oil objective. Quantification of colocalization was performed using the ImageJ JACoP (Just Another Colocalization Plugin[59] version 2.0). Pearson's coefficient is a commonly used colocalization indicators that measures the strength of a linear relationship between two variables[60]. Using JACoP, Pearson's coefficient of GFP and tdTomato signals was calculated for each cell.

## RNA-sequencing and Gene Set Enrichment Analysis

Total RNAs were extracted from day 8 BMDMs (9–10 weeks old male mice: 4 Cre− and 4 Cre+) using the Quick-RNA miniprep plus kit (Zymo). With a minimum of 300 ng input RNA, strand-specific, poly(A)+ libraries were prepared and sequenced at 20 million 100-bp paired-end reads per sample. Raw sequencing reads were mapped to the mouse genome version GRCm39 (M27) using Salmon[61] (version 1.5.1) to obtain transcript abundance counts. MultiQC was used to generate quality control reports based on Salmon read mapping results. The transcript-level count information was summarized to the gene level using tximport[62] (version 1.20.0). Differential expression was assessed using DESeq2[63] (version 1.34.0). Genes with an absolute fold-change > 1.5 and false discovery rate (FDR)-adjusted P value <0.05 were considered as differentially expressed (DE). The output of DESeq2 were scored and ranked based on P value and shrunken $\log_2$ fold-change by apeglm[64] using ranking metrics −log10 P value multiplied by the sign of log-transformed fold-change[65]. The ranked gene list was then used for Gene Set Enrichment Analysis (GSEA)[66] (version 4.2.0) with the weighted statistics to identify the gene sets overrepresented at the top or bottom of the ranked list using the Human Reactome Pathway (the most actively updated general-purpose public database of human pathways) and the Gene Ontology Biological Process annotation (the most commonly used resource for pathway enrichment analysis) within the Molecular Signatures Database. Only ontologies with more than 15 genes and less than 200 genes were considered. g:Orth was used to translate gene identifiers from mouse to human based on the information retrieved from the Ensembl database[67].

## Ingenuity pathway analysis

Ingenuity pathway analysis (IPA) software using build-in scientific literature based database (according to IPA Ingenuity Web Site, www.ingenuity.com) was used to identify canonical pathways, overrepresented in top-scored CRISPR screen hits.

## Quantitative RT-PCR

Total RNA was extracted using Quick-RNA Miniprep Kit (Zymo) and cDNA was synthesized using High-Capacity cDNA Reverse Transcription Kit (Applied Biosystems) as per the manufacturer's instructions. To measure gene expression, quantitative RT-PCR was performed using POWERUP SYBR Green Master Mix by QuantStudio™ 7 Flex Real-Time PCR System (Applied Biosystem, 4485701). ΔΔCT method was used to analyze the relative levels of each transcript normalized to human ACTB.

## In vivo thymus efferocytosis assay

Cre+ and Cre− mice of 8-12 weeks old were injected intraperitoneally with 200 μL PBS or 200 μL PBS containing 250 μg dexamethasone. Dexamethasone was prepared freshly by diluting 4X stock in DMSO with sterile PBS. 18 h after injection, mice were weighed and euthanized, and thymi were harvested and both lobes were weighed. One lobe was immersed in OCT and snap-frozen for immunohistochemical staining to determine efferocytosis in situ, while the other lobe was mechanically disaggregated into single-cell suspension for flow cytometry[25].

To evaluate in situ efferocytosis[25], frozen thymus specimens were cryosectioned at 4-μm and placed on Superfrost plus microscope slides. Sections were fixed in 4% PFA for 10 min and permeabilized in

1% Triton X-100 for 15 min. After rinsing with PBS for three times, sections were incubated with TUNEL staining reagents at 37 °C for 60 min and then washed three times with PBS. Sections were then blocked with 5% goat serum for 60 min at room temperature, followed by overnight incubation at 4 °C in anti-CD68 antibodies (Abcam) diluted in PBS supplemented with 5% BSA to label macrophage area. After washing in PBS, sections were incubated with fluorescently-labeled secondary antibodies and counterstained with DAPI. Images were captured using ImageXpress Micro4 with a Nikon Plan Apo 40X/0.95 objective lens. For quantification, the TUNEL+ nuclei in close proximity or in contact with CD68 + macrophages were counted as macrophage-associated ACs, indicative of efferocytosis. The TUNEL+ nuclei without neighboring macrophages were counted as free ACs. The ratio of macrophage-associated ACs to free ACs was calculated to represent the capability of efferocytosis by thymus macrophages.

To evaluate the percentage of Annexin V+ thymocytes by flow cytometry, mechanically disaggregated thymus cells were rinsed twice with cold DPBS containing 2% HI-FBS and 1 mM EDTA. Cells were then stained with AF647-conjugated Annexin V in Annexin V binding buffer (Invitrogen) at a concentration of $5 \times 10^6$ cells/mL for 15 min at room temperature, followed by flow cytometry analysis.

## In vivo peritoneal macrophage efferocytosis assay

Cre+ and Cre- mice of 12 weeks old were injected intraperitoneally with $1 \times 10^7$ TAMRA-stained apoptotic mouse thymocytes in 300 μl PBS. 15 min after injection, mice were euthanized and peritoneal exudates were collected. The pelleted cells were blocked with CD16/32 (BioLegend) and then stained by FITC-conjugated F4/80 antibody (BioLegend) to label macrophages. The percentage of TAMRA + PMs was determined by flow cytometry[27].

## siRNA-mediated gene silencing and transfection

Non-targeting siRNA and WDFY3-targeting siRNA (Dharmacon) were transfected using Lipofectamine RNAiMAX (Invitrogen) as per the manufacturer's recommendation. Briefly, human PBMCs were seeded at $4 \times 10^5$ per well of 24-well culture dish for differentiation to HMDMs for 7 days with ~70% confluence. HMDMs were then transfected with a final concentration of 25 pmol siRNA and 1 μL Lipofectamine RNAiMAX in 500 μL Opti-MEM (Invitrogen) for 6 h. A second transfection with the same condition was performed 18 h after the completion of the first transfection. HMDMs were collected 48 h from the start of the first transfection for assessing mRNA and protein expression, and efferocytosis capacities.

## Mouse complete blood cell count (CBC) and differential count

Retro-orbital bleeding was performed to collect ~500 μL blood per mouse for complete blood count and differential count using a Heska Element HT5 by the diagnostic lab at the Institute of Comparative Medicine, Columbia University Irvine Medical Center.

## Statistical analyses

Statistical analyses were performed using GraphPad Prism 7. Data were tested for normality using the D'Agostino-Pearson test (when $n >= 8$) or Shapiro-Wilk test (when $n < 8$). Data that passed normality tests were analyzed using Student's $t$ test for comparison of two groups. When $n$ was less than or equal to 5 or when data did not pass normality test, the nonparametric Mann–Whitney test was used. Two-way ANOVA was performed for two independent variables (factors) with two or more groups. Tukey's post hoc test was applied to correct multiple comparisons. Data analyzed using parametric tests were presented as mean ± standard error of mean (SEM), while data analyzed using nonparametric tests were presented as median ± 95% confidence interval (CI). Statistical significance of difference was accepted when P values were <0.05. The specific P values, the number of independent experiments or biological replicates (mice), and the

number of technical replicates per independent experiment and biological replicate were specified in figures and figure legends.

## Reporting summary
Further information on research design is available in the Nature Portfolio Reporting Summary linked to this article.

## Data availability
The datasets generated in this study have been deposited in the Gene Expression Omnibus (GEO), including RNA-seq datasets under accession code GSE211694 and CRISPR screening datasets under accession code GSE212008. The human macrophage RNA-seq dataset was previously published and are available at DRYAD with identifier doi:10.5061/dryad.866t1g1nb. Source data are provided in the Supplementary Information/Supplementary Data/Source Data file. Source data are provided with this paper.

## Code availability
Code used for data analyses is available via the Zhang lab GitHub repository https://github.com/hanruizhang/NatCommun-bioRxiv.2022.477299, and at Zenodo using https://doi.org/10.5281/zenodo.7402413 [https://zenodo.org/record/7402413#.Y46DS7LP1kE].

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

## Acknowledgements

The authors' research work has received funding from R00HL130574 and R01HL151611, and the Irving Scholar award through UL1TR001873 by the National Center for Advancing Translational Sciences (NCATS) and National Institutes of Health (NIH) (to HZ), R00HL145131 (to A.Y.J.), R21MH115347 and grant by Shriners Hospitals for Children (to K.S.Z.), R35HL145228 (to I.T.), R01NS077111 and R01NS101663 (to A.Y.), the Russell Berrie Diabetes Foundation Diabetes Scholar Program (to X.W.), American Heart Association Postdoctoral Fellowships 21POST829654 (to X.W.) and 20POST35130003 (to F.L.), and an NSF predoctoral fellowship (to K.R.C). The content in this manuscript is solely the responsibility of the authors and does not necessarily represent the official views of the NIH. We would like to acknowledge the NIH funding sources to the Columbia Center for Translational Immunology (CCTi) Flow Cytometry Core by grant number S10OD020056 and S10RR027050 and P30DK063608; the NIH-supported microscopy resources in the Center for Biologic Imaging, specifically the confocal microscope supported by grant number 1S10OD019973-01; the NIH/NCI Cancer Center Support Grant P30CA013696 for the use of resources at the Columbia Genome Center; the Columbia Stem Cell Initiative (CSCI) Flow Cytometry Core under the leadership of Michael Kissner; and the High-Throughput Screening Facility at the JP Sulzberger Columbia Genome Center under the leadership of Dr. Charles Karan. FACS cell sorting was performed with great help from Dr. Caisheng Lu, the Technical Director of the CCTi Flow Cytometry Core. We thank Dr. Xiaoli Sky Wu and Dr. Kenneth Chang at the Cold Spring Harbor Laboratory for their technical inputs for CRISPR screening design. We thank Dr. Oren Parnas at the Hebrew University of Jerusalem for his inputs on genome-wide CRISPR screening in primary cells. We thank Dr. Young Joo Yang for technical advice on the characterization of WDFY3. Schematic figures (Figs. 1a, 1g, 2a, 2g, 3b, 5a, 5g, 6a, 7) were created with BioRender.com.

## Author contributions

J.S., X.W., and H.Z. conceived and designed the study. J.S. and X.W. performed majority of the experiments. Z.W., F.L., Y.M., R.M.M., J.C., H.Z. performed experiments. J.S. X.W., Z.W., H.Z. analyzed the data and prepared the figures. J.S. and H.Z. performed CRISPR screening, and the bioinformatic analyses of CRISPR screening and murine RNA-seq data. C.X. performed bioinformatic analyses of human macrophage RNA-seq data. J.S., X.W., and H.Z. wrote the paper with inputs from all authors. K.R.C., A.Y. Jr, J.G.D., W.L. provided guidance for key techniques. K. S. Z. and A.Y. provided key reagents, mice, and critical technical inputs. I.T., A.Y. advised on the project and critically reviewed the paper. H.Z. mentored the performance of the work and supervised the funding.

## Competing interests

The authors declare no competing interests.
