## [Peer Review File · Nature Communications]

A Genome-wide CRISPR Screen Identifies WDFY3 as a Regulator of Macrophage EfferocytosisREVIEWER COMMENTS

Reviewer #1 macrophage efferocytosis (Remarks to the Author):

In this study, the authors identify WDFY3 using a genome-wide CRISPR screen as a positive regulator of efferocytosis by macrophages. They present data, using several different approaches, that WDFY3 affects efferocytosis through modulating actin depolymerization as well as phagosomal acidification. Furthermore, they show that the defective phagosomal acidification in WDFY3 deficient macrophages during efferocytosis is caused by a defect of LC3 recruitment to phagosomes required for LAP. As a molecular mechanism for this, they suggest that WDFY3 is able to regulate LC3 recruitment to phagosomes through interaction with GABARAP. Based on these data, they claim that WDFY3 is a new gene that positively regulates efferocytosis through modulating LAP and actin depolymerization.

Overall, the strength of the study is the unbiased genome-wide screen using Cas9 knock-in mice. The authors well describe the problems of the previous screens to identify a gene modulating efferocytosis and advantages of their own screen. The link of WDFY3 to efferocytosis through LAP is also potentially interesting. However, I have several concerns with the lack of mechanistic data and methodology to support their conclusion.

Major comments

1. The authors show that WDFY3 deficiency causes delayed actin polymerization and less phagosomal acidification during efferocytosis. A molecular mechanism how WDFY3 is involved in the phagosomal acidification is relatively well addressed. Nevertheless, there are no mechanistic data how WDFY3 regulates actin depolymerization. Although there are two major defects in efferocytosis by WDFY3 deficient macrophages, how WDFY3 is involved in modulating actin depolymerization is not explored. At least, the authors are recommended to test the pathway suggested by in the previous study (Ref 14 in the manuscript).

2. The author show data that WDFY3 specifically modulate phagocytosis of apoptotic cells but not other targets such as zymosan, RBC, and IgG opsonized targets. Phagocytosis of various targets including apoptotic cells and pathogens is mediated through LAP. A molecular mechanism by which the effect of WDFY3 is limited to efferocytosis but not phagocytosis of other targets, which are also phagocytosed through LAP, need to be tested/discussed.

3. A crucial process during LAP is recruitment of LC3 to phagosomes, and the authors claim that WDFY3 is involved in LAP through modulating LC3 phagosome recruitment. Thus, showing LC3 association with phagosomes using various approaches is essential. The authors show LC3 phagosome association using westernblotting and flow cytometry which are indirect ways. The authors are recommended to show LC3 phagosome recruitment using microscopy (Fig. 3 and 4). In addition, data for the subcellular localization of WDFY3 before and after apoptotic cell feeding will provide spatial relevance for the role of WDFY3 during efferocytosis through LAP.

4. The authors use very inaccurate way to determine the degree of degradation of engulfed apoptotic cells. Distinction between fragmented and non-fragmented apoptotic cells in macrophages is unclear in Fig. 3a and Fig. 6a. The criteria for fragmented apoptotic cells are vague and subjective. I recommend the authors to use a different method to observe apoptotic cell degradation in phagocytes. For example, the author may use the decay rate of a fluorescent dye, e.g. TAMRA, labeling apoptotic cells in macrophages (PMID: 31792382).

5. In Fig. 5. The number of macrophages and apoptotic cells in Cre⁺ mice is higher than that of macrophages in Cre⁻ mice after injection of dexamethasone, but the thymic cellularity is comparable between Cre⁺ and Cre⁻ mice. Usually, increased apoptotic cells in thymi cause less decreased thymic cellularity after injection of dexamethasone in the mouse model. In addition, the number of macrophages are even higher in Cre⁺ mice than in Cre⁻ mice. These numbers are statistically significant. The authors need to explain how these unusually phenomena occur. The authors also need to have data showing the rate of apoptosis of thymocytes from Cre⁻ and Cre⁺ mice. Then, the efficiency of efferocytosis in the mice can be compared fairly. Furthermore, the number of macrophages is significantly higher in Cre⁺ mice than in Cre⁻ mice after dexamethasone injection (Fig. 5d), but it seems that there are more macrophages in Cre⁻ mice than Cre⁺ mice after dexamethasone injection (Fig. 5f).

Additional comments

1. In Fig. 2h, As the quantified graph, the images are also required to be shown at the time points, 10, 20, 40, and 60 min.
2. In fig. 3a, the label of the Y axis should be checked. It seems that it is not % but relative non-fragmented engulfed AC.
3. In Fig. 3c, the authors are asked to check WDFY3 expression in peritoneal macrophages.
4. In Fig. 3e, co-immunoprecipitated WDFY3 is unclear. A total cell lysate blot for WDFY3, GABARAP and an IP blot for GABARAP will help interpret the data. In addition, please indicate a size marker on the blot (for all westernblots).
5. In Fig. 4b, phagosomal acidification is partially rescued by the WDFY3 fragment although it fully rescues the defect of LC3 phagosome association, which needs to be discussed.
6. In Fig. 4, the fragment rescue experiments need to have more controls. Although the defects of phagosome acidification and LC3 association during efferocytosis by WDFY3-deficient macrophages in Fig. 3, controls for Cre⁻ and Cre⁺ cells transduced with lentivirus without the fragment are required.
7. The authors mention about Fig 4e in the text but there is no Fig 4e in Figure 4, which need to be corrected.
8. In fig. 6e. a scale bar is required.

Reviewer #2 CRISPR screens (Remarks to the Author):

In this manuscript, Shi and colleagues examine the mechanisms underlying efferocytosis an important mechanism for clearance of apoptotic cells. The authors employ a forward genetic screen to identify WDFY3 as a novel, positive regulator of efferocytosis in murine macrophages. Using fluorescent reporters and distinct genetic approaches (eg. two distinct knockout mice) the authors comprehensively characterize the role of WDFY3 in regulating efferocytosis. The authors confirm WDFY3 as being important in human macrophage efferocytosis and use in vivo models of efferocytosis. This study was well designed, and the data largely support the authors' primary conclusions. However, there are some concerns with rigor that would help better support the conclusions drawn and should being addressed.

These concerns do not take away from these exciting data that globally characterize efferocytosis for the first time. These findings are likely to have broad interest in the immunology and cell biology communities.

Major comments

1. In Fig2E The authors conclude that loss of WDFY3 does not affect AC binding at 30 minutes of exposure to ACs. These data are difficult to interpret because the ratio of cells to AC was not explicitly stated. This is important given that the uptake phenotype observed in 2D seems to only occur at high ratios and only becomes robust after long-term incubation (>30 minutes). This needs to be clarified. It also is important to show that binding is not affected at the later 60 minute timepoint at a 5:1 ratio where the authors observed the greatest uptake difference in order to fully support the conclusions drawn.

2. To ensure rigor of the findings the underlying image quantifications should be more explicitly described or re-quantified. It appears that an automated, non-biased method of quantification like cell profiler was not used. This approach would bolster the manuscript and ensure all conclusions are supported by the data.

3. In general, the immunoblots throughout are not to high quality publication standards and I'd encourage the authors to redo them. This is particularly problematic in Figure 3 and 4 with the L3 blots. For Figure 4 the loading controls are not consistent and appear to be overloaded thus the quantification compared to Actin is likely not accurate and the conclusions are thus not supported by these data. In Figure 3F/G the authors conclude that the loss of WDFY3 resulted in a decrease in LC3-II in response to ACs. However, it appears from the data trends in Fig 3f that there is an increase in LC3-II in the Cre⁺ cells with ACs added but one data point is driving the authors conclusion. Additionally, it appears that in the Cre⁺ cells without ACs, there is a potentially decrease in LC3-II at basal levels. This suggests that there could be an equivalent fold change increase in LC3-II between the conditions. These problems must be addressed to support the conclusions.

Minor comments

1. It is not explained what the arrows are depicting in Fig 2f, or why they are different colors.

Reviewer #3 macrophage molecular pathways (Remarks to the Author):

The manuscript of Shi et al. describes the results of a CRISPR knockout screen performed in bone marrow derived macrophages from Cas9 transgene mice, and aimed to identify positive and negative regulators of apoptotic cell uptake (efferocytosis). Using flow cytometry based method discriminated highly efferocytic versus non-efferocytic cells, the authors identify known as well as novel positive and negative regulators of efferocytosis.

The authors focus their investigation on clarifying the role of one positive regulator of efferocytosis WDFY3, a protein previously implicated in autophagy. Using a series of mouse in vitro experiments using macrophages from myeloid-specific WDFY3 knockout mice, the authors show that WDFY3 accelerates efferocytosis by promoting F-actin depolymerization as well as promoting lysosomal degradation of engulfed material likely via LC3-associated phagocytosis. These findings are further supported by data obtained in vivo as well as in human primary macrophages.

The study is very clearly designed and written, the experimental data are very well presented and easy to understand. The experiments are described with sufficient technical details. The study results fully support the study conclusions, and there are no flaws in data or their presentation.

Overall, this is a very good quality manuscript, which without doubt significantly advances

our knowledge on efferocytosis and its mechanisms. Therefore, the study can be published in Nature Communications without any major modifications.

Minor:

Fig 6d – the number of replicates is given as 3, but four points are in the figure.

RESPONSE LETTER

Summary of responses to the Reviewers' comments:

We appreciate the reviewers for the constructive feedback. We have addressed the comments point by point as outlined below. We believe the additional work during the revision has enhanced rigor and transparency further and has provided additional mechanistic insights into how WDFY3 is required for both the uptake and degradation of apoptotic cell cargo by macrophages. Changes in the revised manuscript are highlighted in **green font**. The following data are newly added or modified in the revised manuscript:

- Fig. 2h
- Fig. 3b, 3e, 3f
- Fig. 4a-e
- Fig. 5f
- Fig. 6f
- Supplementary Fig. 5f-h
- Supplementary Fig. 7a-b
- Supplementary Fig. 11a-b
- Supplementary Notes 1-3

Reviewer #1 macrophage efferocytosis (Remarks to the Author):

Overall Comments: *In this study, the authors identify WDFY3 using a genome-wide CRISPR screen as a positive regulator of efferocytosis by macrophages. They present data, using several different approaches, that WDFY3 affects efferocytosis through modulating actin depolymerization as well as phagosomal acidification. Furthermore, they show that the defective phagosomal acidification in WDFY3 deficient macrophages during efferocytosis is caused by a defect of LC3 recruitment to phagosomes required for LAP. As a molecular mechanism for this, they suggest that WDFY3 is able to regulate LC3 recruitment to phagosomes through interaction with GABARAP. Based on these data, they claim that WDFY3 is a new gene that positively regulates efferocytosis through modulating LAP and actin depolymerization. Overall, the strength of the study is the unbiased genome-wide screen using Cas9 knock-in mice. The authors well describe the problems of the previous screens to identify a gene modulating efferocytosis and advantages of their own screen. The link of WDFY3 to efferocytosis through LAP is also potentially interesting. However, I have several concerns with the lack of mechanistic data and methodology to support their conclusion.*

Overall Responses: We thank the reviewer for the insightful feedback and constructive suggestions, and for highlighting the strength of the unbiased screen and the novelty of the discovery of WDFY3. We have provided new data to delineate further how WDFY3 may regulate cargo-specific uptake and to strengthen the methodology. Point-by-point responses are provided below. As we were working on those new experiments, we do also recognize that fully addressing the molecular mechanisms would require extensive amounts of work that will warrant independent studies. Those studies are important and will deepen our understanding of the fundamental mechanisms of cargo-specific processing by macrophages. We have provided our insights into the potential experiments to address the unanswered questions in the Discussion section.

Major comments:

Comment 1: *The authors show that WDFY3 deficiency causes delayed actin polymerization and less phagosomal acidification during efferocytosis. A molecular mechanism how WDFY3 is involved in the phagosomal acidification is relatively well addressed. Nevertheless, there are no mechanistic data how WDFY3 regulates actin depolymerization. Although there are two major defects in efferocytosis by WDFY3 deficient macrophages, how WDFY3 is involved in modulating actin depolymerization is not explored. At least, the authors are recommended to test the pathway suggested by in the previous study (Ref 14 in the manuscript).*

Response 1: We appreciate the reviewer for the insightful comments. Indeed, it is important to understand the molecular mechanisms by which WDFY3 regulates F-actin disassembly, and, relevant to **Comment 2**, how this WDFY3-mediated F-actin dynamics contribute to AC-specific uptake. As the reviewer has pointed out, Ref 14 (PMID: 26465210) revealed that the engulfment of larger cargos (e.g. 5 μ m beads) requires phosphoinositide 3-kinase (PI3K)-mediated PtdIns(3,4,5)₃ production and PtdIns(3,4,5)₃-dependent recruitment of GTPase-activating proteins (GAPs) that inactivates Rac/Cdc42, therefore allowing synchronized F-actin assembly and disassembly. We expect that this mechanism is also required for the engulfment of ACs (~10 μ m for Jurkat cells). Indeed, PI3K inhibitor markedly reduced the uptake of ACs in both control and *Wdfy3* knockout BMDMs, implicating that WDFY3 was not upstream of nor required for PI3K activation. We reasoned that if WDFY3 is entirely downstream of PI3K-mediated F-actin disassembly, with PI3K inhibitor treatment, knockout of *Wdfy3* should not impair AC uptake further. In fact, with PI3K inhibition, *Wdfy3* knockout BMDMs showed lower AC uptake compared with control BMDMs (**Supplementary Fig. 5f**), supporting that WDFY3 affects AC uptake at least partly through PI3K and GAP-independent mechanisms. As expected, when PI3K is inhibited, uptake of 10 μ m beads was comparable between *Wdfy3* knockout and control BMDMs (**Supplementary Fig. 5g**), suggesting that WDFY3-mediated regulatory mechanisms were not required for beads engulfment. Consistently, the percentage of BMDMs with F-actin-ring surrounded beads was also comparable between *Wdfy3* knockout and control BMDMs (**Supplementary Fig. 5h**), in sharp contrast to the higher percentage of F-actin-ring surrounded engulfed ACs in *Wdfy3* knockout BMDMs compared with control BMDMs (**Fig. 2h**), supporting defective F-actin disassembly. The results were described on **Page 6 of the revised manuscript**.

Therefore, our new experiments confirmed that **(1)** WDFY3 was not upstream of nor required for PI3K activation; **(2)** the mechanisms by which WDFY3 regulates F-actin disassembly were at least partly independent of PI3K-mediated GAP recruitment and Rac/Cdc42 inactivation; **(3)** WDFY3-mediated F-actin dynamics was not required for beads engulfment. Although our current work does not yet fully address the molecular mechanisms, the data support the need for extensive experiments to identify WDFY3 binding partners, e.g. using pull-down of different WDFY3 domains followed by mass spectrometry to unbiasedly discover protein binding partners with or without AC engulfment, followed by validation using immunoprecipitation, immunofluorescence staining, and live cell imaging of endogenously tagged WDFY3. These works are important, though to a scale that likely requires multiple independent studies. We are intrigued and committed to conducting these studies as our ongoing and future efforts, as also summarized in the Discussion section to share our insights with the community. We believe our work paves the way for innovative discoveries on the fundamental mechanisms of the regulation of F-actin dynamics that have never been explored before and set the stage for continuing to fully resolve the differential requirement of WDFY3 in cargo-specific uptake during efferocytosis.

Comment 2: *The author show data that WDFY3 specifically modulate phagocytosis of apoptotic cells but not other targets such as zymosan, RBC, and IgG opsonized targets. Phagocytosis of various targets including apoptotic cells and pathogens is mediated through LAP. A molecular mechanism by which the effect of WDFY3 is limited to efferocytosis but not phagocytosis of other targets, which are also phagocytosed through LAP, need to be tested/discussed.*

Response 2: We concur with the reviewer that it is an important observation that WDFY3 specifically modulates uptake of ACs, yet is dispensable for the uptake of other substrates. Despite decades of efforts, how

macrophages involve different molecular machinery to regulate the engulfment of various cargos remains largely undetermined. As discussed in **Response 1**, PI3K-dependent GAP recruitment was found to be essential for the engulfment of large beads by facilitating synchronized F-actin polymerization and depolymerization. The pathway is also involved in AC engulfment, yet our new data support that WDFY3's role in F-actin dynamics was not important for beads engulfment. Because macrophages more effectively engulf rigid cargos (e.g. latex beads) than soft cargos (such as ACs) that deform therefore requiring stronger mechanical force, we speculate that WDFY3-mediated F-actin dynamics is required for the uptake of the more challenging cargos, such as ACs. To our best knowledge, cargo-specific uptake is largely known as being regulated by ligand/receptor binding, while there is very limited understanding of the intracellular molecular machinery determining how macrophages may deal with the more challenging vs. the less challenging cargos. Our works discovered WDFY3 as such a regulator, demonstrating the power of CRISPR screening using ACs as the substrates, and represent the initial efforts to fully address the fundamental mechanisms employed by macrophages to recruit different molecular machinery for cargo-specific uptake. As also outlined in **Response 1**, we envision that extensive amounts of work will be required to fully answer these important questions, and we have highlighted our vision and insights into the questions and potential experiments in the **Discussion section** as the reviewer has suggested.

LC3-associated phagocytosis (LAP) is a process wherein elements of autophagy conjugate LC3 to phagosomal membranes. LAP is indeed required for the degradation of various targets, including ACs and pathogens. The lack of key LAP components, e.g. RUBCN, exclusively affects LC3-dependent phagosome-lysosome fusion and degradation, without altering uptake (PMID: 26098576, Page 895 - "However, Rubicon-/- macrophages were unable to translocate LC3 to LAPosomes, despite equivalent phagocytosis, Fig. 1b-d and Supplementary Fig. 2g"). As the reviewer has also pointed out, the role of WDFY3 in interacting with GABARAP through C-terminal domains and facilitating LC3 lipidation and acidification supports its role in LAP. Yet, WDFY3 has another major role in regulating uptake, also supported by our data that the C-terminal WDFY3 rescued LC3 lipidation, but not uptake defects. We have extended our discussion further to clarify the two major defects in LAP and uptake due to WDFY deficiency on **Page 9 of the revised manuscript**.

Comment 3: *A crucial process during LAP is recruitment of LC3 to phagosomes, and the authors claim that WDFY3 is involved in LAP through modulating LC3 phagosome recruitment. Thus, showing LC3 association with phagosomes using various approaches is essential. The authors show LC3 phagosome association using western blotting and flow cytometry which are indirect ways. The authors are recommended to show LC3 phagosome recruitment using microscopy (Fig. 3 and 4). In addition, data for the subcellular localization of WDFY3 before and after apoptotic cell feeding will provide spatial relevance for the role of WDFY3 during efferocytosis through LAP.*

Response 3: We agree with the reviewer that demonstrating WDFY3 localization and LC3 phagosome recruitment using microscopy is critical to strengthen the conclusion on the role of WDFY in LAP. Because of the lack of a reliable antibody for immunofluorescence staining of WDFY3 and the technical challenge of packaging the full-length WDFY3 cDNA, which is 10.8 kb thus preventing effective transfection or transduction, we fused tdTomato to C-terminal WDFY3 and transfected the construct via electroporation to GFP-LC3 mouse bone marrow-derived macrophages in order to determine: (1) GFP-LC3 phagosome recruitment during efferocytosis; (2) WDFY3 subcellular localization before and after AC feeding; (3) GFP-LC3 and tdTomato-WDFY3 colocalization. As described by our co-author Dr. Ai Yamamoto (PMIDs: 20417604) and also demonstrated in our data, C-terminal WDFY3 was sufficient to rescue LC3 lipidation and acidification. As shown in the new **Fig. 4e**, C-WDFY3 showed cytoplasmic localization. Without AC engulfment, BMDMs from GFP-LC3 mice showed basal levels of LC3 punta. With AC engulfment, LC3 punta showed association with the phagosome. LC3 and C-WDFY3 colocalization also increased in AC-engulfed BMDMs vs. BMDMs without AC uptake. Thus, the imaging data are consistent with Western Blotting and FACS data, supporting the role of WDFY3 in

interacting with LC3 complex and facilitating LC3 lipidation in LAP. The results were described on **Page 9 of the revised manuscript**.

Comment 4: *The authors use very inaccurate way to determine the degree of degradation of engulfed apoptotic cells. Distinction between fragmented and non-fragmented apoptotic cells in macrophages is unclear in Fig. 3a and Fig. 6a. The criteria for fragmented apoptotic cells are vague and subjective. I recommend the authors to use a different method to observe apoptotic cell degradation in phagocytes. For example, the author may use the decay rate of a fluorescent dye, e.g. TAMRA, labeling apoptotic cells in macrophages (PMID: 31792382).*

Response 4: We thank the reviewer for the constructive critique. We have followed the methods described in PMID: 31792382, and have further confirmed impaired AC degradation in *Wdfy3* knockout BMDMs (**Fig. 3b**). Specifically, we labeled ACs with TAMRA, a dye labeling peptides and proteins. We fed TAMRA-labeled ACs to BMDMs for efferocytosis. After one hour, unbound ACs were washed away and BMDMs were either collected for flow cytometry to quantify the TAMRA intensity that represents the baseline, or returned to the incubator for 16 hours to allow degradation. After 16 hours, BMDMs were collected for quantification of TAMRA intensity by flow cytometry. The rate of degradation was then calculated as the decrease in TAMRA intensity divided by the baseline TAMRA intensity. Similar experiments were performed in HMDMs with siRNA-mediated knockdown of WDFY3, knockdown of WDFY3 impaired AC degradation in HMDM (**Fig. 6f**) and consistent with **Fig. 6e** as quantified by imaging. To further improve transparent reporting for data in **Fig 3a** and **Fig 6e**, we have now included **Supplementary Note 3** to describe and illustrate how we have scored non-fragmented vs. fragmented ACs. We agree that the methods for **Fig 3a** and **Fig 6e** involve subjective scoring, yet the scoring was well established (PMID: 28942921 by our co-author's laboratory) and performed blindly, and we think the imaging-based methods allow visualization of fragmentation of engulfed AC corpse, which complement the quantitative methods by FACS (**Fig 3b** and **Fig 6f**). Therefore, we are reporting results from both experiments in the revised manuscript. The results were described on **Page 7 of the revised manuscript**.

Comment 5: *In Fig. 5. The number of macrophages and apoptotic cells in Cre⁺ mice is higher than that of macrophages in Cre⁻ mice after injection of dexamethasone, but the thymic cellularity is comparable between Cre⁺ and Cre⁻ mice. Usually, increased apoptotic cells in thymi cause less decreased thymic cellularity after injection of dexamethasone in the mouse model. In addition, the number of macrophages are even higher in Cre⁺ mice than in Cre⁻ mice. These numbers are statistically significant. The authors need to explain how these unusually phenomena occur. The authors also need to have data showing the rate of apoptosis of thymocytes from Cre⁻ and Cre⁺ mice. Then, the efficiency of efferocytosis in the mice can be compared fairly. Furthermore, the number of macrophages is significantly higher in Cre⁺ mice than in Cre⁻ mice after dexamethasone injection (Fig. 5d), but it seems that there are more macrophages in Cre⁻ mice than Cre⁺ mice after dexamethasone injection (Fig. 5f).*

Response 5: We agree with the reviewer that the data should be further reconciled. We first confirmed that dexamethasone-induced apoptosis rate of thymocytes is comparable between control and *Wdfy3* knockout mice (**Supplementary Fig. 11a**). The results are expected as *Wdfy3* knockout driven by the LysMCre specifically target myeloid cells, while cells accounting for the majority of the thymi should not be directly affected. We also confirmed further that knockout of *Wdfy3* did not alter dexamethasone-induced apoptosis in BMDMs (**Supplementary Fig. 11b**), ruling out the direct effects of *Wdfy3* knockout on macrophage apoptosis. We also observed that when using the same concentration of dexamethasone, thymocytes (80% are T-cells) are markedly more susceptible to apoptosis than macrophages (**Supplementary Fig. 11a-b**). Thus, we confirm that the efficiency of efferocytosis in the mice can be compared fairly in the dexamethasone-treated model. The results were described on **Page 10 of the revised manuscript**.

The number of macrophages in Cre⁺ mice was not lower than Cre⁻ mice in dexamethasone-treated mice, supporting that macrophage filtration into the thymus was not impaired and that the increased % of apoptotic cells in the Cre⁺ thymus was not merely due to the lack of macrophages. We do recognize that for the thymus efferocytosis experiments, reduced clearance is typically associated with increased cellularity. As the reviewer also noted, we speculate that unresolved inflammation in Cre⁺ mice may contribute to the higher monocyte infiltration and macrophage accumulation in the thymus. Thus although the percentage of macrophages engulfing ACs could be lower, the net outcome for AC clearance may be partly rescued when macrophages number is higher. In addition, impaired clearance and unresolved inflammation ultimately lead to increased cell death, which will affect cellularity counts. We acknowledge that the thymus efferocytosis model assesses macrophage efferocytosis indirectly by readouts on the percentage of ACs. Importantly, our complementary PM *in vivo* efferocytosis assay directly assessed efferocytosis of PM *in vivo* (**Fig. 5h-i**). Therefore, with the complementary approaches, we believe we have demonstrated the role of WDFY3 in macrophage efferocytosis *in vivo*.

For **Fig. 5f**, we have selected another set of representative images to better illustrate the quantified graph.

Minor Comments:

Comment 6: *In Fig. 2h, As the quantified graph, the images are also required to be shown at the time points, 10, 20, 40, and 60 min.*

Response 6: We agree and have updated **Fig. 2h** to include representative images for all time points.

Comment 7: *In fig. 3a, the label of the Y axis should be checked. It seems that it is not % but relative non-fragmented engulfed AC.*

Response 7: Thanks for pointing out the inconsistency. We have updated Fig. 3a to show the % of non-fragmented engulfed ACs.

Comment 8: *In Fig. 3c, the authors are asked to check WDFY3 expression in peritoneal macrophages.*

Response 8: We have now provided the Western Blot data to demonstrate the successful knockout of WDFY3 in PM as shown in the newly added **Supplementary Fig. 7a**.

Comment 9: *In Fig. 3e, co-immunoprecipitated WDFY3 is unclear. A total cell lysate blot for WDFY3, GABARAP and an IP blot for GABARAP will help interpret the data. In addition, please indicate a size marker on the blot (for all westernblots).*

Response 9: We have now included both input and IP blots in **Fig 3e**. We have also provided a size marker for all the blots with the original blots provided in the source data spreadsheet. In addition, we have included detailed protocols and procedures for the IP in the Methods section. Because WDFY3 is ~400 kDa and GABARAP is ~15 kDa, their Western Blotting is intrinsically challenging. We have diligently optimized the protocols in order to obtain the improved blots now shown in the updated **Fig. 3e**. We hope the detailed methods we have included in the Methods section can also benefit the field working on those proteins.

Comment 10: *In Fig. 4b, phagosomal acidification is partially rescued by the WDFY3 fragment although it fully rescues the defect of LC3 phagosome association, which needs to be discussed.*

Response 10: We appreciate the insights and agree that this should be more explicitly discussed. Indeed, we reason that *Wdfy3* knockout led to impaired acidification through two mechanisms: (1) WDFY3 directly interacts with GABARAP/LC3 complex thus facilitating LC3 lipidation and phagosome-lysosome fusion and subsequent acidification; (2) Knockout of *Wdfy3* led to defects in F-actin disassembly, which is expected to delay the subsequent phagosome-lysosome fusion and lysosomal acidification. Therefore, though the C-WDFY3 completely rescued LC3 lipidation, the acidification was not completely rescued, likely because C-WDFY3 was not sufficient to rescue defects in uptake due to impaired F-actin disassembly. The discussion is also provided on **Page 9 of the revised manuscript**.

Comment 11: *In Fig. 4, the fragment rescue experiments need to have more controls. Although the defects of phagosome acidification and LC3 association during efferocytosis by WDFY3-deficient macrophages in Fig. 3, controls for Cre- and Cre+ cells transduced with lentivirus without the fragment are required.*

Response 11: We thank the reviewer for the constructive suggestion. We have now included Cre- and Cre+ cells transduced with lentivirus without the C-WDFY3 for data shown in **Fig. 4**. The controls allowed us to draw an important conclusion that overexpression of C-WDFY3 in Cre-cells did not further enhance uptake, acidification, or LC3 lipidation.

Comment 12: *The authors mention about Fig 4e in the text but there is no Fig 4e in Figure 4, which need to be corrected.*

Response 12: We thank the reviewer for the careful review. The figure number should be **4d** and we have fixed the error.

Comment 13: *In fig. 6e. a scale bar is required.*

Response 13: We have double-checked to include scale bars for all image data. Since all the images in the same panel have the same magnification, only one scale bar is included for images within the same panel.

Reviewer #2 CRISPR screens (Remarks to the Author):

Overall Comments: *In this manuscript, Shi and colleagues examine the mechanisms underlying efferocytosis an important mechanism for clearance of apoptotic cells. The authors employ a forward genetic screen to identify WDFY3 as a novel, positive regulator of efferocytosis in murine macrophages. Using fluorescent reporters and distinct genetic approaches (eg. two distinct knockout mice) the authors comprehensively characterize the role of WDFY3 in regulating efferocytosis. The authors confirm WDFY3 as being important in human macrophage efferocytosis and use in vivo models of efferocytosis. This study was well designed, and the data largely support the authors' primary conclusions. However, there are some concerns with rigor that would help better support the conclusions drawn and should being addressed. These concerns do not take away from these exciting data that globally characterize efferocytosis for the first time. These findings are likely to have broad interest in the immunology and cell biology communities.*

Overall Responses: We appreciate the reviewer for the constructive critique and also for recognizing the broad impact of our research. We have addressed the comments point-by-point as outlined below.

Major Comments:

Comment 1: *In Fig2E The authors conclude that loss of WDFY3 does not affect AC binding at 30 minutes of exposure to ACs. These data are difficult to interpret because the ratio of cells to AC was not explicitly stated. This is important given that the uptake phenotype observed in 2D seems to only occur at high ratios and only becomes robust after long-term incubation (>30 minutes). This needs to be clarified. It also is important to show that binding is not affected at the later 60 minute timepoint at a 5:1 ratio where the authors observed the greatest uptake difference in order to fully support the conclusions drawn.*

Response 1: We thank the reviewer for the comments on methodological details to improve the clarity of our description. In a 2D system, the apoptotic cells (ACs) fed to the macrophage culture will first settle to the bottom of the culture vessel, followed by the “smelling” phase for macrophages to locate ACs, and then the ligand-receptor recognition and binding phase, and subsequent initiation of internalization. There are some heterogeneities, but in general, it takes ~15-30 minutes for the internalization phase to initiate. Therefore, the 30 min timepoint better resembles the binding phase, and therefore was chosen to assess binding. We have also clarified in the text that a 5:1 ratio was used when determining binding capacity. The method (30 minutes with inhibition of actin polymerization either by cytochalasin D or incubation at 4 °C) is widely used by the macrophage biology community to study the binding of AC cargo with macrophages, as shown in the few references listed below and **cited in our revised manuscript (Page 5, Line 178):**

Moon et al., Nat Commun, 2020, Page 6, Figure 2a: <https://www.nature.com/articles/s41467-020-19272-0>
Perry et al., Nat Cell Biology, 2019, Page 1539, Figure 6f: <https://www.nature.com/articles/s41556-019-0431-1>
Wang et al., Cell, 2017, Page 5, Figure 3B: <https://pubmed.ncbi.nlm.nih.gov/28942921/>

When using a higher AC : BMDM ratio or longer co-incubation time, the higher burden challenged the control BMDMs to engulf with maximal capacity. The defects in *Wdfy3* knockout BMDM are therefore further revealed, as shown in **Fig. 2c** and **2d**, explaining why the phenotypic difference becomes robust at high ratios and after long-term incubation.

Comment 2: *To ensure rigor of the findings the underlying image quantifications should be more explicitly described or re-quantified. It appears that an automated, non-biased method of quantification like cell profiler was not used. This approach would bolster the manuscript and ensure all conclusions are supported by the data.*

Response 2: We thank the reviewer for this important comment. Unbiased and automated imaging quantification is powerful, and we apply this analysis strategy as much as possible in our research. In particular, the validation of screening hits in *in vitro* efferocytosis assay was performed using the Nikon Ti-S Automated Inverted Microscope with NIS-Elements High Content Analysis Imaging Software. We have described the analysis (**on Page 17 of the revised manuscript**), and included the analysis template in our Github repository together with other codes for transparent reporting. The colocalization in **Fig. 4e** was analyzed by Image J using the JACoP plugin (**on Page 21 of the revised manuscript**).

We fully agree that quantitative imaging analysis software, including NIS-Elements and CellProfiler, are powerful. Yet, for some of our analysis needs, automated analysis pipelines are yet to be de novo developed and validated before can be reliably applied. We believe this is the challenge for the broad community that requires extensive efforts to continuously improve, and semi-automated and manual scoring remain as important strategies. We do

recognize the limitation of relying on manual analysis and therefore took the following strategy to enhance rigor and transparency:

- For the imaging-based degradation assay in **Fig. 3a** and **6e**, we also performed new experiments using flow cytometry-based quantification of degradation rate as shown in **Fig. 3b** and **6f**, which showed consistent results with the imaging-based assay.
- For the quantification of F-actin ring in **Fig. 2h**, we also had flow cytometry-based analysis in **Fig. 2g** supporting consistent results.
- For assays that need to rely on imaging-based analyses, we have ensured that **(1)** the analysis was performed blindly; **(2)** a fully automated microscope was used to randomly capture multiple fields-of-view per technical replicate (instead of manual select the field-of-view for capturing).
- To ensure rigor and transparency further, we followed the reviewer's suggestion to more explicitly describe the quantification strategy in the newly provided **Supplementary Notes 1-3** for examples and processes of binding assay quantification, F-actin ring quantification, and degradation quantification in BMDM and HMDM. We have also included additional methodological details for quantification in the Methods section.

Comment 3: *In general, the immunoblots throughout are not to high quality publication standards and I'd encourage the authors to redo them. This is particularly problematic in Figure 3 and 4 with the L3 blots. For Figure 4 the loading controls are not consistent and appear to be overloaded thus the quantification compared to Actin is likely not accurate and the conclusions are thus not supported by these data. In Figure 3F/G the authors conclude that the loss of WDFY3 resulted in a decrease in LC3-II in response to ACs. However, it appears from the data trends in Fig 3f that there is an increase in LC3-II in the Cre+ cells with ACs added but one data point is driving the authors conclusion. Additionally, it appears that in the Cre+ cells without ACs, there is a potentially decrease in LC3-II at basal levels. This suggests that there could be an equivalent fold change increase in LC3-II between the conditions. These problems must be addressed to support the conclusions.*

Response 3: We concur with the reviewer that it is important to provide high-quality blots to strengthen the conclusion. We have repeated all the Western Blotting experiments and generated new **Fig. 3e-f** and the entire **Fig. 4**, with higher quality blots and increased sample size for quantification. We confirmed that in **Fig 3f**. Cre- and Cre+ BMDMs showed similar LC3-II. AC increased LC3-II in Cre-, but not in Cre+ BMDMs. Please note that the **Fig. 3f** and **Fig. 4c** blots for ACTB had non-specific bands marked with "*". This is because we were using the same membrane to blot both ACTB and LC3-II, the approach for more reliable normalization and quantification. When blotting ACTB using separate membranes, non-specific bands do not appear, and we confirm the location of ACTB bands as specific signals.

Minor Comments:

Comment 1: *It is not explained what the arrows are depicting in Fig 2f, or why they are different colors.*

Response 1: We thank the reviewer for this comment to improve the clarity of our visualization. We recognized that there are no need to use different colors, and we have also updated the figure legends indicating that the white arrows point to the BMDM engulfing an AC across the stage from phagocytic cup formation to completed phagosome formation.

Reviewer #3 macrophage molecular pathways (Remarks to the Author):

Overall Comments: *The manuscript of Shi et al. describes the results of a CRISPR knockout screen performed in bone marrow derived macrophages from Cas9 transgene mice, and aimed to identify positive and negative regulators of apoptotic cell uptake (efferocytosis). Using flow cytometry based method discriminated highly efferocytic versus non-efferocytic cells, the authors identify known as well as novel positive and negative regulators of efferocytosis. The authors focus their investigation on clarifying the role of one positive regulator of efferocytosis WDFY3, a protein previously implicated in aggrephagy. Using a series of mouse in vitro experiments using macrophages from myeloid-specific WDFY3 knockout mice, the authors show that WDFY3 accelerates efferocytosis by promoting F-actin depolymerization as well as promoting lysosomal degradation of engulfed material likely via LC3-associated phagocytosis. These findings are further supported by data obtained in vivo as well as in human primary macrophages.*

The study is very clearly designed and written, the experimental data are very well presented and easy to understand. The experiments are described with sufficient technical details. The study results fully support the study conclusions, and there are no flaws in data or their presentation. Overall, this is a very good quality manuscript, which without doubt significantly advances our knowledge on efferocytosis and its mechanisms. Therefore, the study can be published in Nature Communications without any major modifications.

Overall Responses: We sincerely appreciate the reviewer for the positive feedback and for highlighting the significance of our work.

Minor Comments:

Comment 1: *Fig 6d – the number of replicates is given as 3, but four points are in the figure.*

Response 1: Thanks to the reviewer for the careful review. We have corrected the error.

REVIEWERS' COMMENTS

Reviewer #1 (Remarks to the Author):

The authors have performed additional experiments to address the points raised in the first round of review. I believe that the revision apparently improves the rigor and clarity of the manuscript and further supports their conclusions. Although a mechanism by which WDFY3 is involved in F-actin dynamics is not fully address, their insights about it is sufficiently discussed. I have no further suggestions for changes and think that the study deserve publication.

Reviewer #2 (Remarks to the Author):

In this revised manuscript the authors describe a genome-wide screen to identify regulators of efferocytosis. They find a new mechanisms related to WDFY3 as a key regulator. The authors have addressed my previous concerns and I believe this manuscript will have a broad impact on the field.

RESPONSE LETTER

Reviewer #1 (Remarks to the Author):

The authors have performed additional experiments to address the points raised in the first round of review. I believe that the revision apparently improves the rigor and clarity of the manuscript and further supports their conclusions. Although a mechanism by which WDFY3 is involved in F-actin dynamics is not fully addressed, their insights about it are sufficiently discussed. I have no further suggestions for changes and think that the study deserves publication.

Responses: We appreciate the reviewer for the constructive feedback that has greatly improved our manuscript.

Reviewer #2 (Remarks to the Author):

In this revised manuscript the authors describe a genome-wide screen to identify regulators of efferocytosis. They find new mechanisms related to WDFY3 as a key regulator. The authors have addressed my previous concerns, and I believe this manuscript will have a broad impact on the field.

Responses: We thank the reviewer for the insightful comments and for recognizing the impact of our work on the field.